# ATP burst is the dominant driver of antibiotic lethality in *Mycobacterium smegmatis*

Tejan Lodhiya[1], Aseem Palande[1], Anjali Veeram[1], Gerald J Larrouy-Maumus[2], Dany JV Beste[3], Raju Mukherjee[1]*

[1]Department of Biology, Indian Institute of Science Education and Research (IISER), Tirupati, India; [2]Centre for Bacterial Resistance Biology, Department of Life Sciences, Faculty of Natural Sciences, Imperial College London, London, United Kingdom; [3]Department of Microbial Sciences, Faculty of Health and Medical Sciences, University of Surrey, Guildford, United Kingdom

## eLife Assessment

In this **important** work, Lodhiya et al. provide evidence that excessive ATP underlies the killing of the model organism *Mycobacterium smegmatis* by two mechanistically-distinct antibiotics. The data are generally **solid** as the authors deploy multiple, orthogonal readouts and methods for manipulating reactive oxygen species and ATP. The work will be of interest to those studying antibiotic mechanisms of action.

*For correspondence:
raju.mukherjee@iisertirupati.ac.in

Competing interest: The authors declare that no competing interests exist.

**Abstract** Antibiotic-tolerant bacteria, due to their unique physiology, are refractory to antimicrobial killing and pose challenges for infection control. Incomplete knowledge of how bactericidal antibiotics work limits our understanding of partial resistance due to phenotypic tolerance in mycobacteria, a driver for developing genetic resistance. Using proteomics, $^{13}$C isotopomer analysis, genetic and biochemical assays, we investigated the physiological response of *M. smegmatis* challenged with aminoglycoside and fluoroquinolone antibiotics. Two distinct classes of antibiotics elicited remarkably similar responses and increased flux through the TCA cycle, causing enhanced respiration, ROS generation, and ATP burst. We observed that excessive ATP levels and not ROS dominantly contribute to cidality, which may in part be conferred by sequestration of divalent metal ions by ATP. Consequently, $^{13}$C isotope tracing indicated TCA cycle flux deviation from its oxidative arm as a bacterial adaptive mechanism, which also included activated intrinsic resistance and a higher propensity to develop antibiotic resistance. Our study provides a new understanding of the intricate mechanisms of antibiotic-induced cell death and expands the current paradigm for antibiotic action.

## Introduction

Antibiotics have saved millions of lives from bacterial infections; however, the emergence of resistance against all classes of antibiotics renders clinically available antibiotics ineffective and undermines efforts to combat lethal bacterial infection. Antimicrobial resistance (AMR) is responsible for approximately 700,000 deaths per year and, if left unchecked, is projected to exceed 10 million by 2050 (*Tagliabue and Rappuoli, 2018*). The conventional approach for combating AMR through pathogen-specific antibiotics is no longer viable, as the drug discovery pipeline has shrunk over the past several decades. Therefore, there is a pressing need to develop novel interventions for preserving and enhancing the

efficacy of the existing drugs through synergy or adjuvant therapy. However, there are gaps in our understanding of how precisely the existing antibiotics mediate cell death and what the bacterial counter responses to these challenges are at a system-wide scale.

Antibiotics are commonly believed to cause cell death either by inhibiting essential cellular processes facilitating cell division or compromising the integrity of the cell envelope, while antibiotic resistance involves more complex mechanisms and manifests as genotypic resistance, antibiotic tolerance, and persistence. A proposed hypothesis posited a unified mechanism underlying antibiotic lethality driven by the presence of toxic reactive oxygen species (ROS), rather than solely targeting their intended primary targets. This hypothesis suggests that antibiotic-induced cell death occurs through a series of interconnected events, including enhanced respiration, disruption of iron-sulfur clusters, increase in intracellular iron levels, and triggering of the Fenton reaction (*Kohanski et al., 2007*; *Lobritz et al., 2015*; *Dwyer et al., 2014*). These processes culminate in the autocatalytic generation of ROS, ultimately leading to oxidation of biomolecules and cell death. However, this hypothesis has faced significant rebuttals. Some studies have shown a lack of ROS production (*Liu and Imlay, 2013*) or no clear correlation between ROS levels and antibiotic lethality (*Keren et al., 2013*). Additionally, research has demonstrated that increased respiration and intracellular iron pool enhance antibiotic lethality not because they produce ROS, but by promoting antibiotic uptake (*Ezraty et al., 2013*). Intriguingly, a recent study examining *E. coli* treated with aminoglycosides revealed that lethality stems directly from dysregulated membrane potential, which was independent of the voltage-dependent drug uptake (*Bruni and Kralj, 2020*) as was widely understood (*Ezraty et al., 2013*; *Davis et al., 1986*; *Taber et al., 1987*). Likewise, the role of active growth rate and metabolic activity in antibiotic efficacy has been subject to contradictory findings (*Greulich et al., 2015*; *Lopatkin et al., 2019*; *Lee et al., 2018*; *Amor and Gore, 2022*; *Lapińska et al., 2022*). Interestingly, respiration-decelerating antibiotics were found to inhibit the lethality of respiration-accelerating antibiotics (*Lobritz et al., 2015*), highlighting the need for a new understanding of the drug's mode of action, particularly in a combination therapy. Despite the intuitive deleterious effects of ROS on bacterial physiology, sub-lethal concentrations were found to promote multidrug resistance by inducing superoxide radical-induced mutations (*Kohanski et al., 2010*). These results demonstrate the intricate nature of the repercussions stemming from antibiotic action and the metabolic factors influencing cell death. They underscore the necessity of investigating antibiotic actions to dissect the specific metabolic factors governing cell death amidst the complex interplay with factors like respiration, ROS, membrane potential, and ATP. Bacterial responses to these mechanisms contribute to the emergence of phenotypic and eventually genetic resistance.

Antibiotic tolerance results from physiological adaptation that enables a bacterial population to withstand lethal doses of antibiotics for a longer duration and also provides the driving force to develop genetic resistance (*Levin-Reisman et al., 2017*). Although the significance of antibiotic tolerance is recognised, the precise metabolic state and sequence of the adaptive events leading to the development of antibiotic tolerance remain poorly characterised. *M. tuberculosis*, the etiological agent for tuberculosis, a prominent cause of death from infectious disease, shows exceptional tolerance to antibiotics and partly explains the reason for a long treatment regimen requiring a combination of antibiotics (*World Health Organization, 2022*). Despite having low rates of mutation and recombination, the rapid increase in multi- and extensively drug-resistant tuberculosis infections compels us to re-examine our current understanding of the mechanisms of antibiotic action in mycobacteria and identify vulnerable targets for adjuvant therapy to shorten the treatment time in TB.

A systemic investigation of the physiological responses to antibiotic exposure prior to the emergence of resistance holds the key to deciphering the nature and quantum of antibiotic-induced stresses. In this study, we employed temporal quantitative proteomics and $^{13}$C isotopomer analysis to comprehensively understand the physiological responses of the model organism, *M. smegmatis*, exposed to sub-lethal concentrations of antibiotics from two distinct classes, fluoroquinolone and aminoglycoside. Through genetic and biochemical assays, we show the involvement of ROS and ATP burst in antibiotic lethality. Here, we attempted to underscore the interplay between ROS and ATP burst in contributing to cell death. Our findings contribute to a deeper understanding of the complex mechanisms underpinning antibiotic-induced cell death in mycobacterium with implications for preventing the development of antibiotic tolerance and resistance.

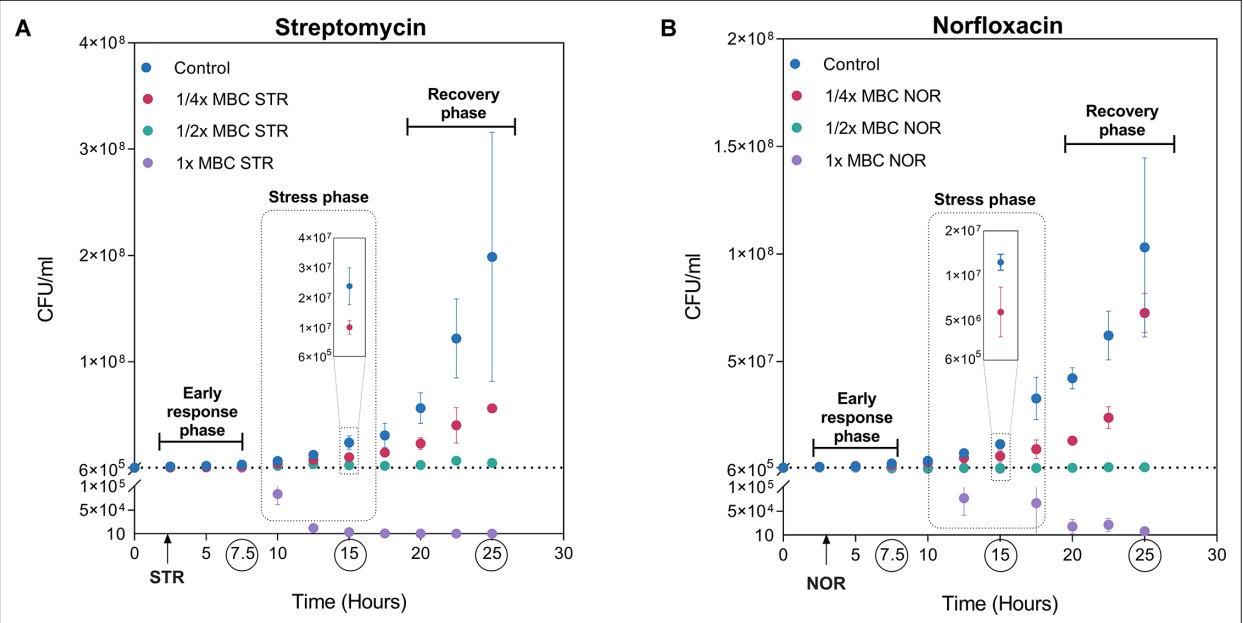

**Figure 1.** Exposure to sub-lethal doses of aminoglycoside and fluoroquinolone reveals physiological adaptation to antibiotics. The growth curve of *M. smegmatis* challenged with either 1 x, ½x, or ¼x MBC$_{99}$ of streptomycin (**A**) or with norfloxacin (**B**). Data points represent the mean of at least three independent replicates ± SD. 1 x MBC$_{99}$ of streptomycin (250 ng/ml) and 1 x MBC$_{99}$ of norfloxacin (2 μg/ml). Bacterial viability at the start was 600,000 CFU/ml, corresponding to an OD$_{600nm}$ of 0.0025 /ml.

The online version of this article includes the following figure supplement(s) for figure 1:

**Figure supplement 1.** MIC verification of drug treated and untreated *M. smegmatis*.

## Results

### Exposure to sub-lethal doses of aminoglycoside and fluoroquinolone reveals physiological adaptation without genetic resistance

In order to understand the biology of antibiotic-stressed mycobacteria, we designed a growth assay to measure the responses of mycobacteria to sublethal antibiotic concentrations. *M. smegmatis* mc$^2$155 (OD$_{600nm}$ = 0.0025) was challenged with either 1 x, ½x, or ¼x MBC$_{99}$ (minimum bactericidal concentration) of fluoroquinolone (norfloxacin) and aminoglycoside (streptomycin) antibiotics. While 1 x MBC$_{99}$ and ½x MBC$_{99}$ concentrations were lethal and growth-inhibitory, respectively, growth at sub-lethal concentrations (i.e. ¼x MBC$_{99}$) of both antibiotics exhibited a prolonged lag phase (*Figure 1*).

To decipher the antibiotic-induced growth changes, we divided the growth curve into three phases: an early response phase (5 hr post-treatment), which is a period shortly after the addition of antibiotics and shows no visible differences in the growth rate; a stress phase (5–15 hr post-treatment), characterised by reduced growth rate and prolonged lag time; and a recovery phase (15–22.5 hr post-treatment), in which mycobacterial growth rate resumes following antibiotic adaptation. The experiments were terminated when the culture reached stationary phase (~ OD$_{600nm}$=1) to prevent the stationary phase response from interfering with the antibiotic-induced response. The prolonged lag time observed during the stress phase is of profound significance in deciphering the nature of antibiotic-induced stresses and has implications for the development of antibiotic tolerance (*Moreno-Gámez et al., 2020*). Similarly, a thorough understanding of the recovery phase cells can reveal cellular determinants involved in adaptation to antibiotics. Towards this end, we first ruled out whether the resumption of growth on sub-lethal concentrations of antibiotics was due to the emergence of an antibiotic-resistant population. For this, bacteria were harvested from the recovery phase (25 hr), and the MIC of the antibiotics on cells grown with and without ¼x MBC$_{99}$ of the same antibiotic were compared. No changes in MIC confirmed that the growth resumption during the recovery phase was physiological rather than a result of genetic resistance (*Figure 1—figure supplement 1*).

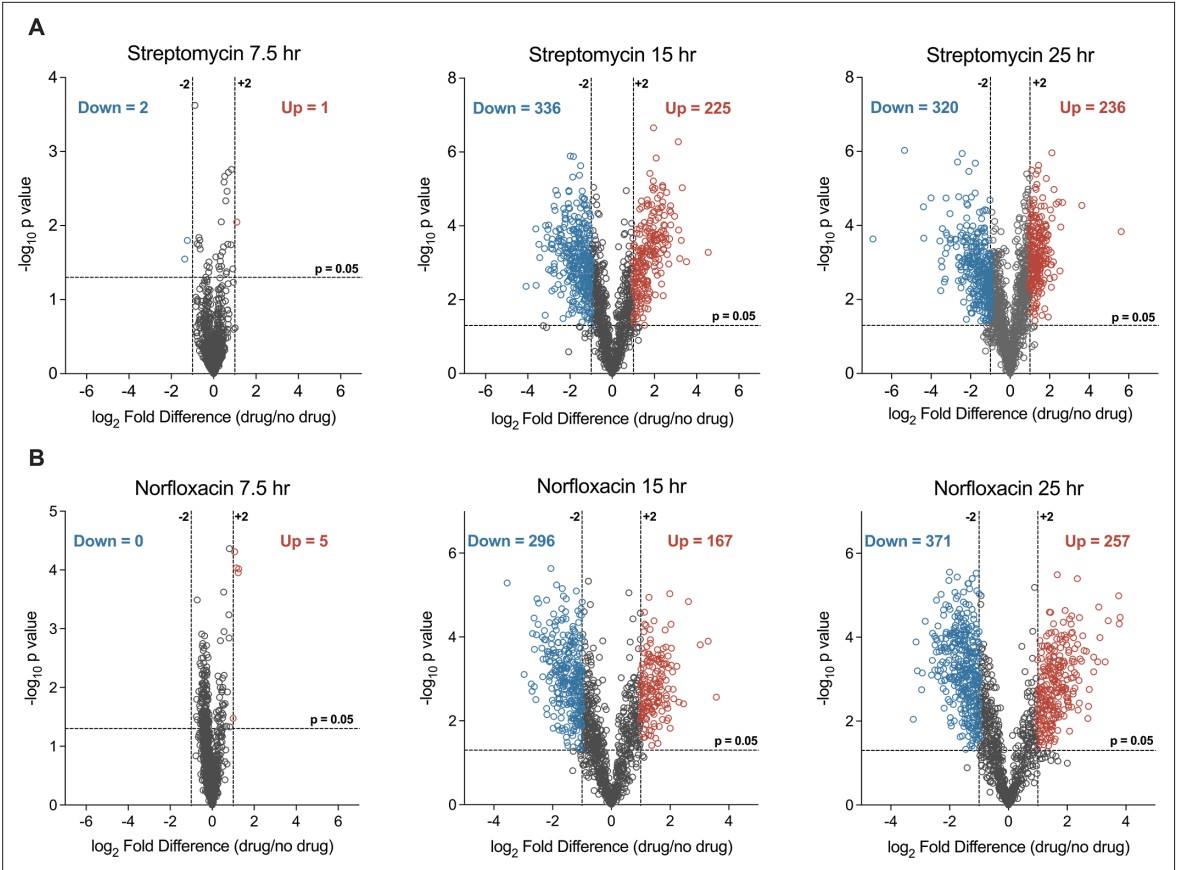

**Figure 2.** Differential expression analysis of total proteome upon norfloxacin and streptomycin treatment. Volcano plot analysis of the proteins temporally quantified at 7.5 hr, 15 hr, and 25 hr time points for streptomycin (**A**) norfloxacin treatment (**B**). Each dot represents one protein. Fold change cut-off ±2 and a t-test significance cut-off of ≤0.05 were applied to identify differentially expressed proteins, shown in different colours.

The online version of this article includes the following source data and figure supplement(s) for figure 2:

**Figure supplement 1.** Cytosolic proteins identified upon treatment with antibiotics.

**Figure supplement 1—source data 1.** Normalized intensities of the protein IDs.

**Figure supplement 1—source data 2.** Volcano plot analysis of proteomics data.

## Norfloxacin and streptomycin induce similar proteomic responses

We performed untargeted label-free quantitative (LFQ) proteomics on mycobacteria grown with and without ¼x MBC$_{99}$ of both antibiotics, during all three growth phases to observe the physiological alterations in mycobacterium exposed to antibiotics. Six datasets yielded a total of 2982 distinct proteins (43% of the *M. smegmatis* proteome), with each sample identifying roughly 2000 proteins (*Figure 2—figure supplement 1A*), thus demonstrating the robustness of our methodology. To ascertain the reproducibility across samples, the data for each condition were subjected to rigorous quality control analysis (*Figure 2—figure supplement 1B*). Differentially expressed proteins were defined as those crossing a fold change threshold of ±2 and a p-value of ≤0.05. A volcano plot analysis revealed the differential expression of several proteins in response to both antibiotics (*Figure 2*). A complete list of proteins identified (i.e. Maxquant protein group file) and differentially expressed (i.e. volcano plot analysis file and list of exclusively expressed proteins) in each six datasets is available as *Figure 2—figure supplement 1—source data 1 and 2*, respectively.

Corresponding to the unaffected growth rate in the early response phase (7.5 hr), the proteomic profile of antibiotic-treated cells was indistinguishable from that of the control group at this time point. In contrast, cells from the stress phase (15 hr) and the recovery phase (25 hr) displayed several up- and down-regulated proteins in response to both antibiotics. We also noted a transition from a coordinated to an uncoordinated state of the proteome, and back to an unperturbed state depending on the

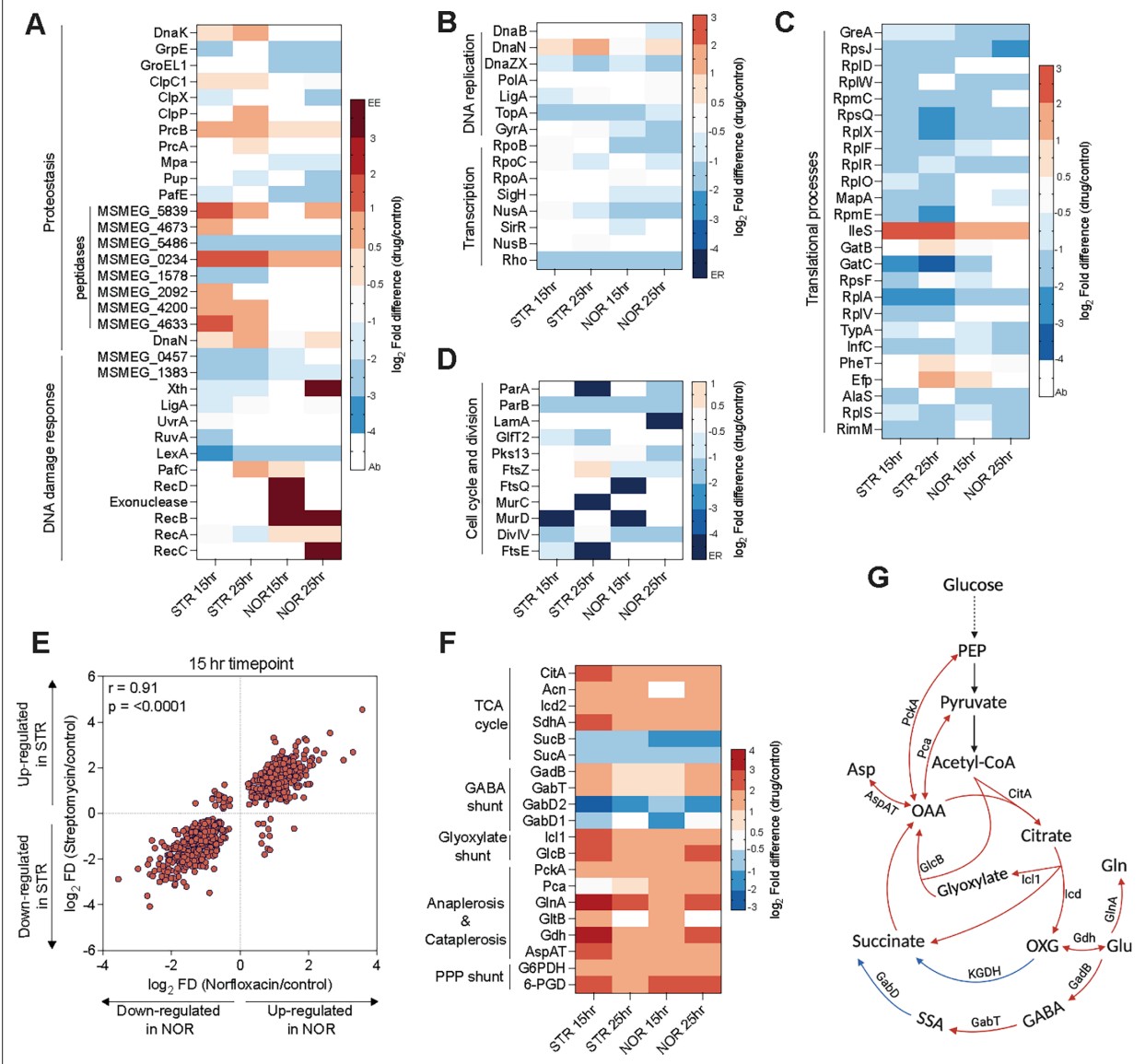

**Figure 3.** Norfloxacin and streptomycin have a common mechanism of action. Heat map analysis represents the log$_2$ fold changes of proteins involved in (**A**) DNA damage response and proteostasis network, (**B**) DNA replication and transcription, (**C**) translation, and (**D**) cell cycle and cell division processes, identified from proteomics data. Colours in the heat map depict the extent of differential expression, where EE demonstrates exclusively expressed, ER demonstrates exclusively repressed/absent in response to antibiotics, and Ab represents proteins that were not identified or quantified with high significance. (**E**) Scatter plot of the Pearson correlation analysis of common proteins identified upon norfloxacin (NOR) and streptomycin (STR) treatment. Each dot represents a single protein with its log$_2$ fold difference for NOR (x-axis) and STR (y-axis) treatment at t=15 hr time point. (**F**) Heat map depicting the log$_2$ fold changes of proteins involved in central carbon metabolism (CCM) of *M. smegmatis*. Panel (**G**) illustrates the CCM pathway of *M. smegmatis* and denotes the expression status of indicated enzymes with colour coding similar to (**F**).

The online version of this article includes the following figure supplement(s) for figure 3:

**Figure supplement 1.** Commonalities in altered proteome upon antibiotic treatment.

time for growth recovery. Consistent with its mode of action, streptomycin (STR) treatment up-regulated proteases, peptidases, and proteasomal complex components implicated in the degradation of misfolded proteins (*Figure 3A*). Likewise, norfloxacin (NOR) treatment induced the SOS response by down-regulating LexA, the repressor of SOS response, and up-regulating DNA repair proteins, such as RecA/B/C and PafC (*Figure 3A*). These data indicate that the sub-lethal doses of antibiotics used in our study exert sufficient stress on their primary targets. Next, we sought to determine the secondary consequences of antibiotic-target interactions. Towards this end, proteins involved in central dogma

(DNA replication, transcription, and translation), and cell cycle and cell division processes were found significantly down-regulated in response to both the antibiotics (*Figure 3B and D*), correlating with the observed decrease in growth rate during the stress phase. Our results suggest that the initial bacterial response to antibiotics may involve downregulation of essential cellular processes that are required for optimal growth and antibiotic lethality.

Next, to test whether NOR and STR share secondary mechanisms of action (*Kohanski et al., 2007*; *Dwyer et al., 2014*), we performed a Pearson correlation analysis between the NOR and STR induced alteration in proteome from the stress (*Figure 3E*) and recovery phases (*Figure 3—figure supplement 1A*). The results revealed a highly correlated (Pearson coefficient (r)>0.90) proteomic profile of antibiotic-treated mycobacteria in the stress phase (*Figure 3E*), indicating that structurally and functionally distinct antibiotics may produce a strikingly similar response. It remains uncertain whether the common response to antibiotics indicates a shared mechanism of action or a general stress response independent of specific modes of action of NOR and STR. To determine the cause of these similarities, we attempted to gain a deeper insight into the shared processes impacted by antibiotics. For this, we conducted pathway enrichment analysis on proteins that were up-regulated and exclusively expressed upon antibiotic treatments. As depicted in *Figure 3—figure supplement 1B*, pathway enrichment analysis showed that central carbon metabolic pathways were enriched for both antibiotics. Notably, enzymes of the TCA cycle (CitA, Icd, Acn), glyoxylate shunt (Icl1 and GlcB), GABA shunt, the anaplerotic node (Pck, Pca), and PPP shunt (G6-PDH, 6-PGD) were significantly up-regulated for both antibiotics, suggesting metabolic alterations in response to antibiotic treatment (*Figure 3F and G*). Collectively, these data suggest a common mechanism of antibiotic action that extends beyond their primary targets.

## Norfloxacin and streptomycin generate reactive oxygen species to confer cidality

Studies have reported that ROS are generated as a secondary consequence of antibiotic-target interaction and facilitate cell death by oxidizing vital biomolecules (*Dwyer et al., 2014*). We sought to analyse the expression status of proteins involved in anti-oxidant response to check whether the antibiotics under study generate ROS. Antibiotic-treated proteome revealed a significant upregulation of anti-oxidant enzymes, such as catalases, superoxide dismutases, mycothione reductase, and methionine sulfoxide reductase (MsrA) (*Figure 4A*). The upregulation of enzyme MsrA, involved in the repair of oxidised proteins, indicates ROS-mediated protein oxidation. In addition, anti-oxidant proteins, such as alkylhydroxyperoxidase, bromoperoxidase, and peroxiredoxin were exclusively expressed in response to antibiotics, suggesting that bacteria may have experienced oxidative stress upon antibiotic treatment. Pathway enrichment analysis (*Figure 3—figure supplement 1B*) revealed upregulation of cysteine and methionine metabolic pathways for both the antibiotics, which form a central component of the redox homeostasis in Mtb. Cysteine contributes to the synthesis of mycobacterial antioxidant, mycothiol (*Buchmeier et al., 2003*), donates sulphur to generate Fe-S clusters (*Rybniker et al., 2014*), and is required for respiration and antibiotic tolerance (*Mishra et al., 2019*; *Schnappinger et al., 2003*). Together, the upregulation of cysteine and antioxidant pathways suggests that the treatment with NOR and STR induces ROS production and causes oxidative stress.

In order to directly measure oxidative stress (due to intracellular ROS production) in response to antibiotics, we used the Mrx1-roGFP2 redox biosensor (*Bhaskar et al., 2014*). It is a ratiometric fluorescence reporter that measures the intracellular redox potential of mycobacteria and has been extensively used to measure the redox state of cells in vitro and ex vivo (*Bhaskar et al., 2014*; *Saito et al., 2021*; *Shee et al., 2022*). The sensor exhibits an increased fluorescence ratio at 405/488 nm excitation under oxidative stress and a reduced ratio under reductive stress. The ability of the biosensor to measure oxidative and reductive stress was confirmed using an oxidizing and a reducing agent in comparison to the untreated control (*Figure 4—figure supplement 1A*), demonstrating its reliability in sensing oxidative stress in mycobacteria. Thereafter, to interrogate whether the antibiotics induce oxidative stress, we administered sub-lethal to lethal doses of both antibiotics to Mycobacteria containing the biosensor. Exposure to either NOR or STR increased the fluorescence ratio at 405/488 nm, suggesting generation of ROS in a time (*Figure 4B and C*) and concentration-dependent manner (*Figure 4D and G*). Furthermore, antibiotic-induced ROS was directly detected using the CellROX deep red dye. A dose-dependent increase in the mean fluorescence intensity was observed

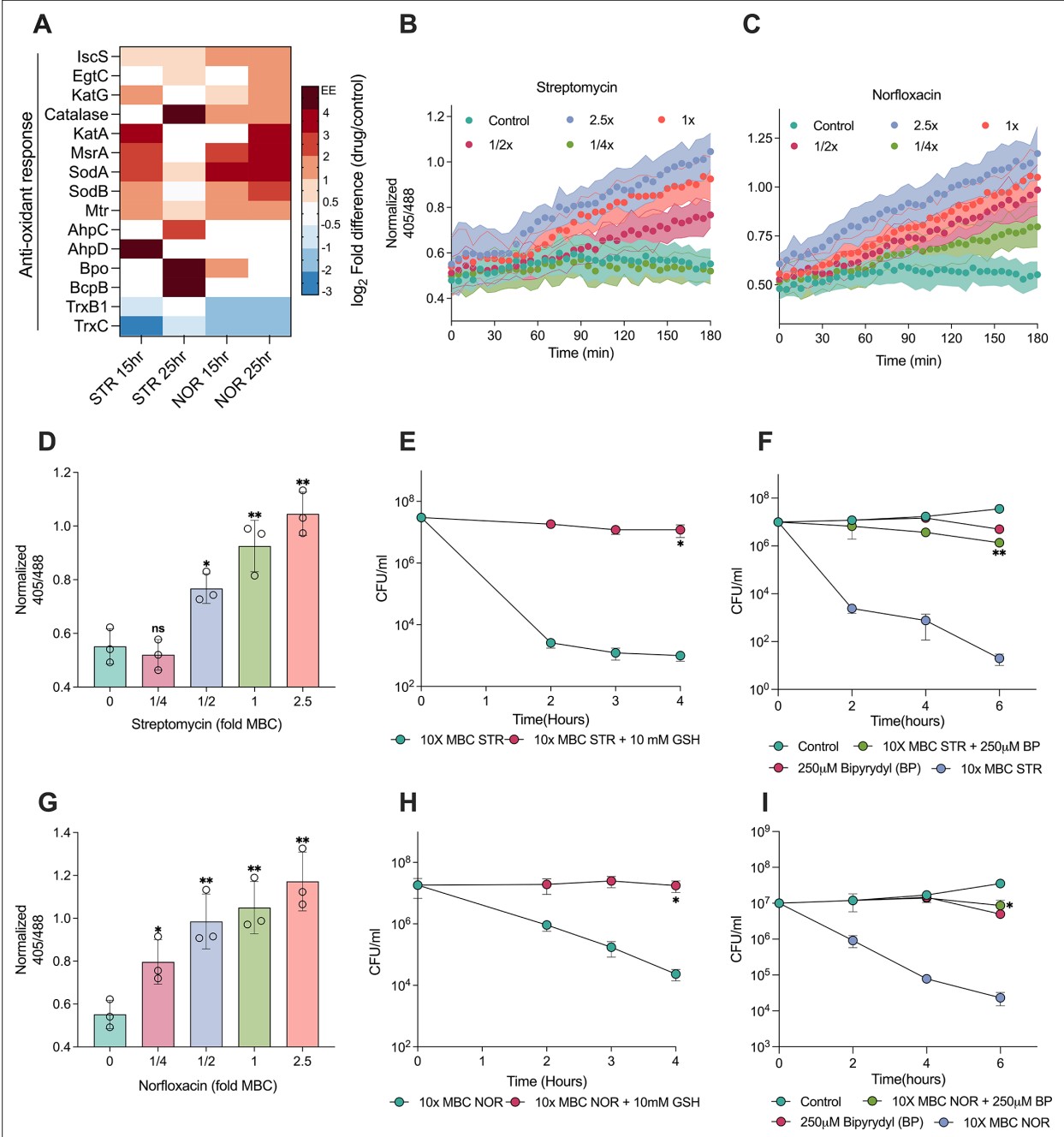

**Figure 4.** Norfloxacin and streptomycin generate reactive oxygen species as part of their lethality. The heat map (**A**) depicts the $\log_2$ fold differences of proteins involved in antioxidant response. EE demonstrates proteins that are exclusively expressed in response to norfloxacin (NOR) or streptomycin (STR). Temporal (**B, C**) and dose-dependent (**D, G**) ratiometric response of *M. smegmatis* expressing Mrx1-roGFP2 redox biosensor upon STR and NOR treatment at 3 hr. The biosensor fluorescence measurements were recorded at the excitation of 405 nm and 488 nm, for a common emission at 520 nm. Panels demonstrate the effect of 10 mM glutathione on the survival of *M. smegmatis* treated with lethal dose (10 x $MBC_{99}$) of STR (**E**) and NOR (**H**). Panels (**F**) and (**I**) present the effect of bipyridyl on the survival of *M. smegmatis* treated with lethal doses of STR and NOR, respectively. Data points represent the mean of at least three independent replicates ± SD. Statistical significance was calculated by Student' t-test (unpaired), \*$p<0.05$, \*\*$p<0.01$, \*\*\*$p<0.001$.

The online version of this article includes the following figure supplement(s) for figure 4:

**Figure supplement 1.** Ratio metric response of *M. smegmatis* expressing Mrx1-roGFP2.

in the case of NOR (**Figure 4—figure supplement 1B**), which, however, was not significant upon STR treatment, presumably due to the selectivity of CellROX dye to detect only the superoxide ($O_2^{\bullet-}$) ions (**McBee et al., 2017**), whose levels may have been lower in the case of STR.

Next, we sought to investigate whether ROS contributed to antibiotic lethality. Towards this end, we first established whether glutathione (GSH), an antioxidant, can quench antibiotic-induced ROS. Indeed, co-treatment with GSH prevented an increase in the 405/488 nm ratio upon antibiotic treatment (**Figure 4—figure supplement 1C and D**). To check the impact of ROS on cell survival, an antibiotic time-kill curve was performed with a supra-lethal dose (10x-$MBC_{99}$) of the antibiotics over a duration predetermined to result in the death of >99.9% of cells (**Balaban et al., 2019**), in the absence and presence of GSH. As observed in **Figure 4E and H**, co-treatment with GSH at non-lethal concentration (**Figure 4—figure supplement 1E**) substantially inhibited the lethality of antibiotics, demonstrating that antibiotic-induced ROS contributed to cell death.

ROS is known to damage nearly all macromolecules, including DNA, proteins, and lipids. Proteins of TCA cycle enzymes and the ETC chain incorporate Fe-S clusters for electron exchange that are very susceptible to ROS-induced damages, resulting in the release of iron ($Fe^{+2}$) in the cytoplasm and induction of the Fenton cycle. Free hydroxyl radicals are generated via the Fenton cycle, which results in further ROS production and could also contribute to antibiotic-induced cell death (**Vilchèze et al., 2013**). Using the iron-chelator bipyridyl (BP), known to inhibit the Fenton cycle-mediated burst of ROS, we found that indeed, iron sequestration significantly reduced antibiotic-induced death (**Figure 4F and I**). Collectively, the induction of the antioxidant response, the detection of ROS using the Mrx1-roGFP2 biosensor, and the mitigation of antibiotic lethality using glutathione and Fenton cycle inhibitor confirm that both NOR and STR treatment cause increased ROS production as a part of their lethality.

## Norfloxacin and streptomycin treatment induces a burst in ATP levels

Despite knowing the significance of ROS generation in bacterial physiology, our understanding of its origin, precise cause, and the stimulus remains unclear. Since the electron transport chain (ETC) is the primary source of ROS production by electron leakage, we hypothesised that the increase in ROS levels resulted from enhanced respiration. To test this, we measured the levels of ATP, the end product of aerobic respiration, in response to antibiotics using the BacTiter-Glo kit. Expectedly, NOR and STR treatments induced a time- and dose-dependent increase in ATP production, suggesting an increase in respiration (**Figure 5A–F**). We next used an orthogonal method to substantiate our results of elevated ATP levels observed upon antibiotic treatment. The PHR-mCherry is a ratiometric fluorescence biosensor that reports ATP/ADP ratio in real time in mycobacteria (**Akela and Kumar, 2021**). Consistent with previous ATP measurement, data presented in **Figure 5G**, demonstrate a dose-dependent increase in ATP/ADP ratio upon NOR and STR treatments, indicating an increase in oxidative phosphorylation and ATP production upon antibiotic treatments.

Of note, both of these methods have an inherent limitation. While in growth-based assays, luminescence can be normalised to $OD_{600nm}$ or total protein content, experiments involving bacterial death upon antibiotic treatment, employing a similar normalisation could lead to incorrect measurement of ATP levels per bacterium. Hence, we opted to normalize the data using viable counts as reported elsewhere (**Gengenbacher et al., 2010**). However, a recent study revealed that antibiotic-induced ROS production may persist even after drug removal, potentially preventing bacteria from forming colonies post-treatment (**Hong et al., 2019**), resulting in underestimating viability and reporting higher RLU/CFU values. Furthermore, the ratiometric biosensor has a limited dynamic range, and its effectiveness in measuring ATP may not be optimal during oxidative damage and cell death. Therefore, it was not feasible to determine the exact magnitude of the increase in ATP levels subsequent to antibiotic exposure. Notably, previous studies reporting a surge in ATP levels using $OD_{600nm}$-based normalisation yielded only 4–5-fold increase upon antibiotic treatment that resulted in over 3–4 $\log_{10}$ fold killing (**Shetty and Dick, 2018**; **Lee et al., 2019a**). Since dead cells can be assumed to be metabolically incapable of contributing to active ATP synthesis, normalisation with viable cells would have generated a similar increase in ATP levels (RLU/CFU) even in slow-growing *M. bovis* BCG and Mtb.

To further test whether enhanced respiration caused elevated ATP levels, we employed the redox dye resazurin to measure bacterial respiratory or metabolic activity upon antibiotic treatments, as reported earlier (**Schrader et al., 2021**). Resazurin reduction assay showed enhanced fluorescence

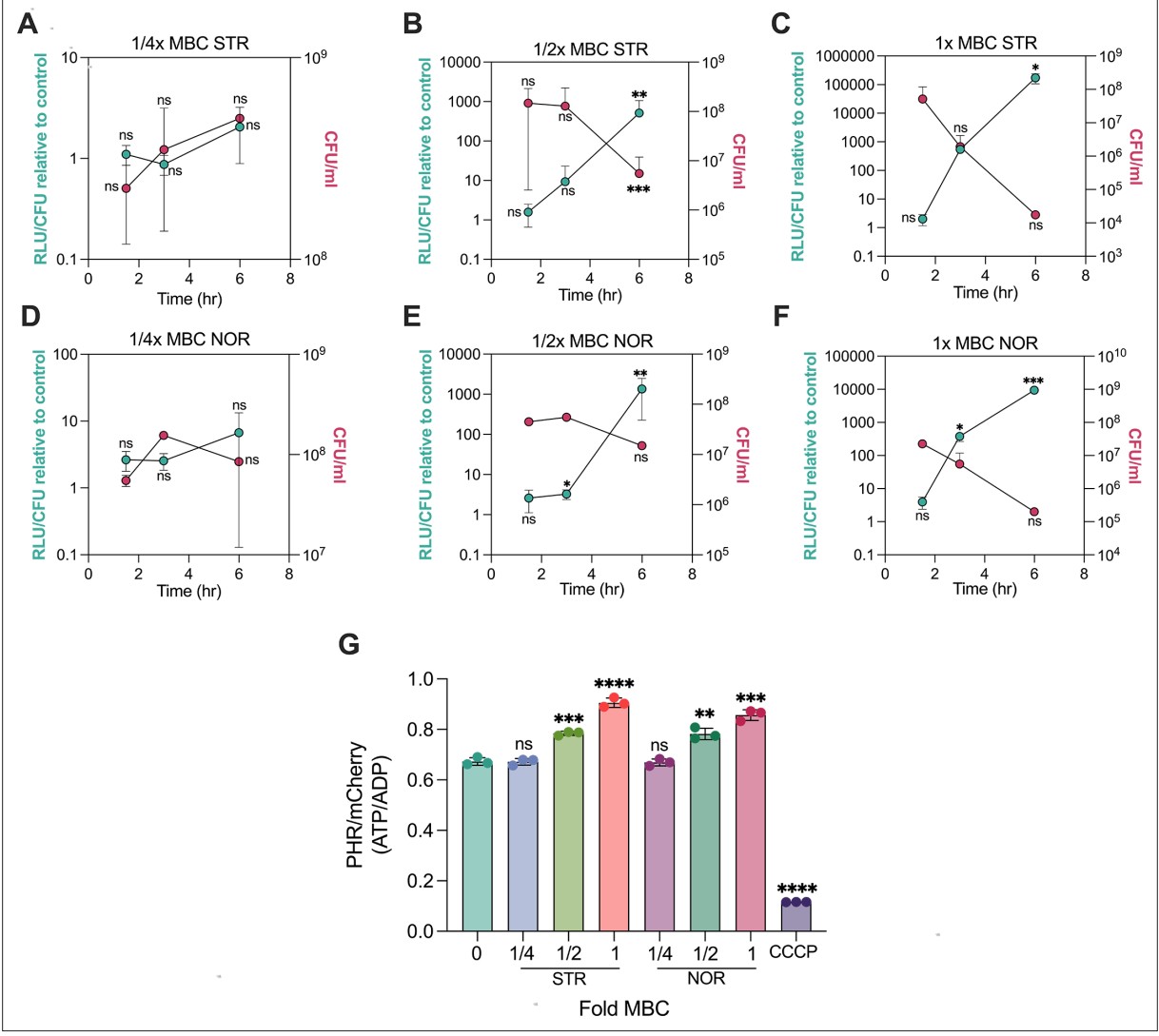

**Figure 5.** Norfloxacin and streptomycin treatment induces a lethal burst in ATP levels. Time course of relative luminescence measurements representing ATP levels in *M. smegmatis* in response to ¼x (**A, D**), ½x (**B, E**), 1 x $MBC_{99}$ (**C, F**) of streptomycin and norfloxacin, respectively. The right y-axis represents the viability of cells measured just before the measurement of ATP. Bar graph (**G**) depicting PHR/mCherry ratios (ATP/ADP) of the ATP biosensor in *M. smegmatis* exposed to increasing concentrations of NOR and STR for 3 hr; 50 micromolar CCCP was used as a control. 1 x $MBC_{99}$ for streptomycin (STR) and norfloxacin (NOR) are 1 µg/ml and 16 µg/ml, respectively for $OD_{600}$=0.8. All data points represent the mean of at least three independent replicates ± SD. Statistical significance was calculated by Student' t-test (unpaired), *$p<0.05$, **$p<0.01$, ***$p<0.001$, ****$p<0$.

The online version of this article includes the following figure supplement(s) for figure 5:

**Figure supplement 1.** Norfloxacin and streptomycin induce metabolic activity in *M. smegmatis*.

conversion rate in bacteria treated with antibiotics, thus confirming increased respiration rate (*Figure 5—figure supplement 1*). Despite over 3–4 $log_{10}$ fold killing observed with 1 x MBC drugs at 6 hr, resazurin reduced faster in antibiotic-treated cells than in untreated cells. Intriguingly, the observed rise in ATP levels as well as in resazurin conversion rate upon antibiotic treatment were inversely proportional to cell survival (*Figure 5A–F*, *Figure 5—figure supplement 1*). Thus, our results suggest that antibiotic-treated cells exhibit a significant increase in respiration, resulting in excessive ATP production that is associated with bacterial death.

## Elevated ATP levels contribute to antibiotic cidality

We next sought to decipher whether the elevated ATP production was a consequence or a cause of antibiotic lethality. To establish the contribution of increased ATP levels to mycobacterial death, it was

necessary to specifically inhibit the antibiotic-induced ATP burst without reducing the overall metabolic rate. To achieve this, we co-treated cells with antibiotics and carbonyl cyanide 3-chlorophenylhydrazone (CCCP), a protonophore, at non-toxic levels to mitigate ATP burst (*Figure 6—figure supplement 1*). CCCP carries protons (ions + charge) across the membrane, thereby collapsing the pH gradient (ΔpH) and the membrane potential ($\Delta\phi$), resulting in disruption of ATP synthesis from oxidative phosphorylation. Thus, CCCP uncouples oxidative phosphorylation from ETC, without directly affecting ATP synthase or the respiratory chain. Expectedly, co-treatment with CCCP dissipated the antibiotic-induced ATP burst in a dose-dependent manner (*Figure 6A, B, D and E*), confirming that the increase in ATP levels mediated by NOR and STR treatment was the result of increased respiration followed by oxidative phosphorylation. Subsequently, we reasoned that if ATP burst contributes directly to antibiotic action, then dissipating antibiotic-induced ATP burst would increase bacterial survival on antibiotics. To test this hypothesis, we compared antibiotic kill curves with CCCP (mitigating ATP burst) and without CCCP (causing ATP burst). As observed in *Figure 6C and F*, co-treatment with CCCP significantly inhibited antibiotic-induced cell death, demonstrating the role of ATP burst in antibiotic lethality.

Debatably, the uptake of aminoglycosides is thought to be affected by the electrochemical gradient (ΔpH and $\Delta\phi$) (*Ezraty et al., 2013*; *Bruni and Kralj, 2020*; *Taber et al., 1987*). Since CCCP disrupts both ΔpH and $\Delta\phi$, it collapses the PMF across the membrane (*Nicholls, 2013*) and may inhibit the uptake of streptomycin. Therefore, to rule out the possibility of reduced streptomycin uptake being responsible for the increased survival in the presence of CCCP, we measured ATP levels and cell survival in the presence of nigericin. Nigericin is a $K^+/H^+$ antiporter and carries out electro-neutral exchange of $H^+$ only to collapse ΔpH, leaving the $\Delta\phi$ unaffected and the PMF compensated (*Nicholls, 2013*). Consistent with the results obtained with CCCP, nigericin co-treatment prevented antibiotic-induced increase in ATP levels (*Figure 6—figure supplement 2A and B*) and consequently rescued mycobacteria against lethal levels of STR or NOR (*Figure 6—figure supplement 2C*). The involvement of ATP in antibiotic action was further confirmed using bedaquiline (BDQ), an inhibitor of mycobacterial ATP synthase, to suppress antibiotic-induced ATP burst. BDQ has been shown to act by collapsing ΔpH without affecting $\Delta\phi$ in *M. smegmatis*, making it ideal for our experiments involving STR (*Hards et al., 2015*). BDQ is known to have a delayed effect on *M. smegmatis* ATP levels, with 100 x MIC of BDQ reducing cellular ATP levels by only 10-fold after 2 days of treatment (*Lu et al., 2014*). Therefore, we sought to give a pre-treatment of BDQ before challenging cells with NOR and STR. While 24 hr of pre-treatment with BDQ at 25 x to 200x-MIC$_{99}$ had no effect on cell survival, they significantly reduced the killing rate of STR (*Figure 6—figure supplement 2D*) and NOR (*Figure 6—figure supplement 2E*).

To ascertain our findings further, we made use of the genetic method to demonstrate the involvement of ATP in antibiotic action to rule out off-target activity of CCCP, nigericin, and BDQ. The *M. smegmatis* Δ*atpD* strain lacks one of the two copies of the *atpD* gene encoding the β subunit of the ATP synthase and has diminished levels of the β subunit, causing respiratory slowdown and lower ATP levels (*Patil and Jain, 2019*). Due to its inability to synthesise ATP equivalent to that of wild-type cells, we hypothesised that Δ*atpD M. smegmatis* would be unable to generate an ATP burst at wild-type scale, upon antibiotic treatment. Indeed, the Δ*atpD* strain displayed lower ATP levels (*Figure 6G*) and enhanced survival (*Figure 6H and I*) upon antibiotic treatment, confirming the significant contribution of ATP burst towards antibiotic lethality. Notably, the growth profile of the Δ*atpD* strain was indistinguishable from wild-type *M. smegmatis* (*Figure 6—figure supplement 3*), indicating that the Δ*atpD* strain had similar growth and metabolic rate to wild-type cells during the exponential phase of the growth and that the reduction of neither contributed to the enhanced survival observed upon antibiotic treatment. Together, these data show that mitigation of antibiotic-induced ATP burst using chemical (CCCP, nigericin, and BDQ) or genetic (Δ*atpD M. smegmatis*) methods diminishes antibiotic lethality, and therefore, elevated ATP levels are metabolic drivers of antibiotic lethality. The causal link between increased ATP levels and antibiotic action observed in our study aligns with previous findings that strongly link low ATP levels to antibiotic persistence (*Conlon et al., 2016*; *Shan et al., 2017*), and high metabolic activity with antibiotic lethality (*Kohanski et al., 2007*; *Lopatkin et al., 2019*; *Kitzenberg et al., 2022*; *Lobritz et al., 2015*; *Lobritz et al., 2022*; *Yang et al., 2019*). Furthermore, our study suggests ATP as one of the pivotal metabolic factors influencing the outcome of antibiotic action and persistence within the broader category of metabolic activity.

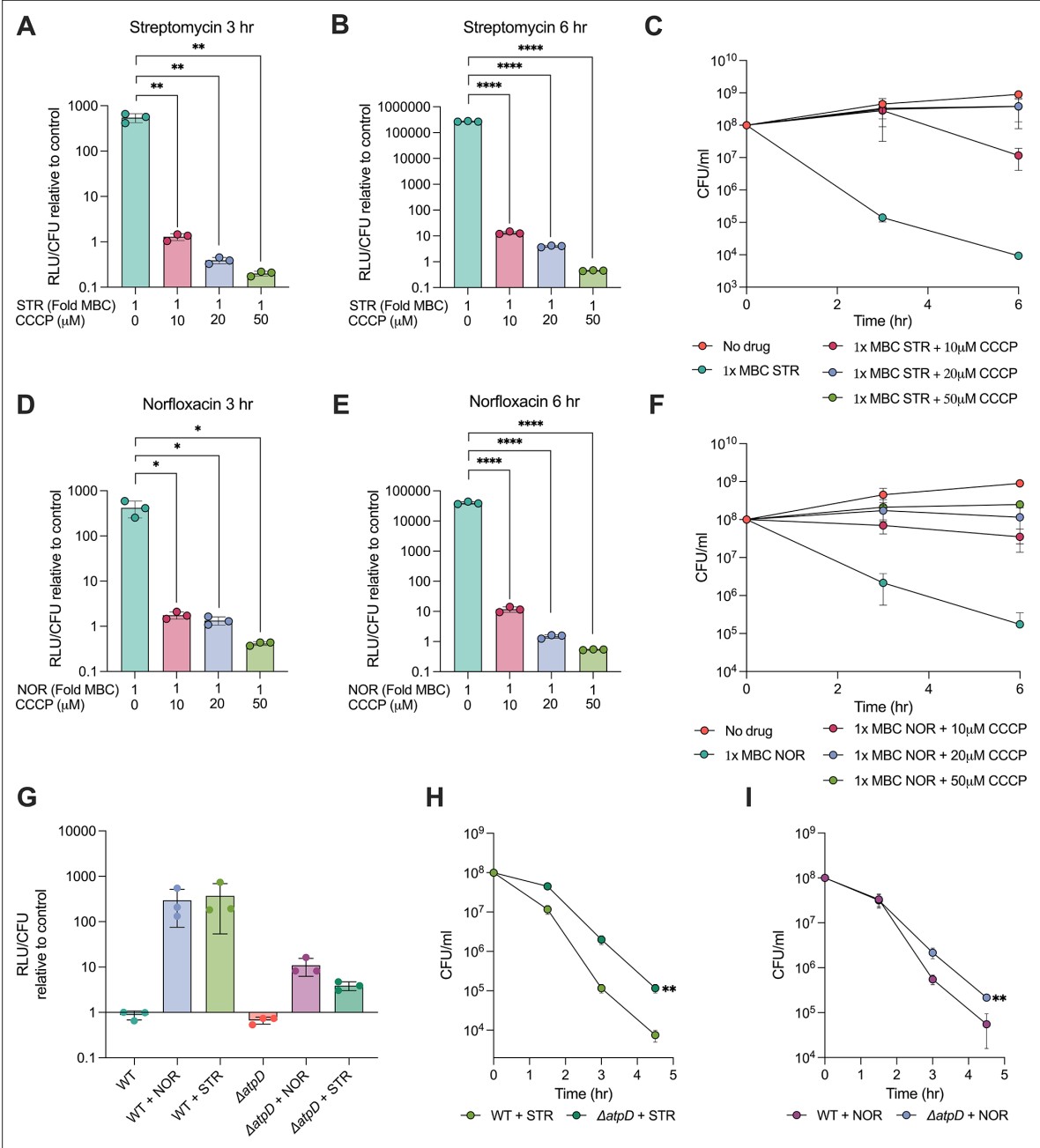

**Figure 6.** Inhibition of excess ATP production mitigates antibiotic lethality. Bar graphs report the dose-dependent reduction in ATP levels upon carbonyl cyanide 3-chlorophenylhydrazone (CCCP) co-treatment with streptomycin (STR) (**A, B**) and norfloxacin (NOR) (**D, E**). The effect of increasing concentrations of CCCP on survival of *M. smegmatis* challenged with lethal doses of STR (**C**) and NOR (**F**), respectively. Bar graphs represent the relative luminescence measurements indicating ATP levels in wild-type and Δ*atpD* in response to 1 x MBC$_{99}$ of STR and NOR for 3 hr (**G**). Time-kill curves demonstrate the survival differences of wild-type and Δ*atpD* in response to 1 x MBC$_{99}$ of STR (**H**) and NOR (**I**). All data points represent the mean of at least three independent replicates ± SD. Statistical significance was calculated by Student' t-test (unpaired), *$p<0.05$, **$p<0.01$, ***$p<0.001$, ****$p<0$.

The online version of this article includes the following figure supplement(s) for figure 6:

**Figure supplement 1.** The effect of CCCP uncoupler on ATP levels and survival of *M. smegmatis*.

**Figure supplement 2.** Suppression of ATP burst by Bedaquiline and Nigericin mitigates antibiotic lethality.

**Figure supplement 3.** The growth curve of wild-type and Δ*atpD* strain of *M. smegmatis*.

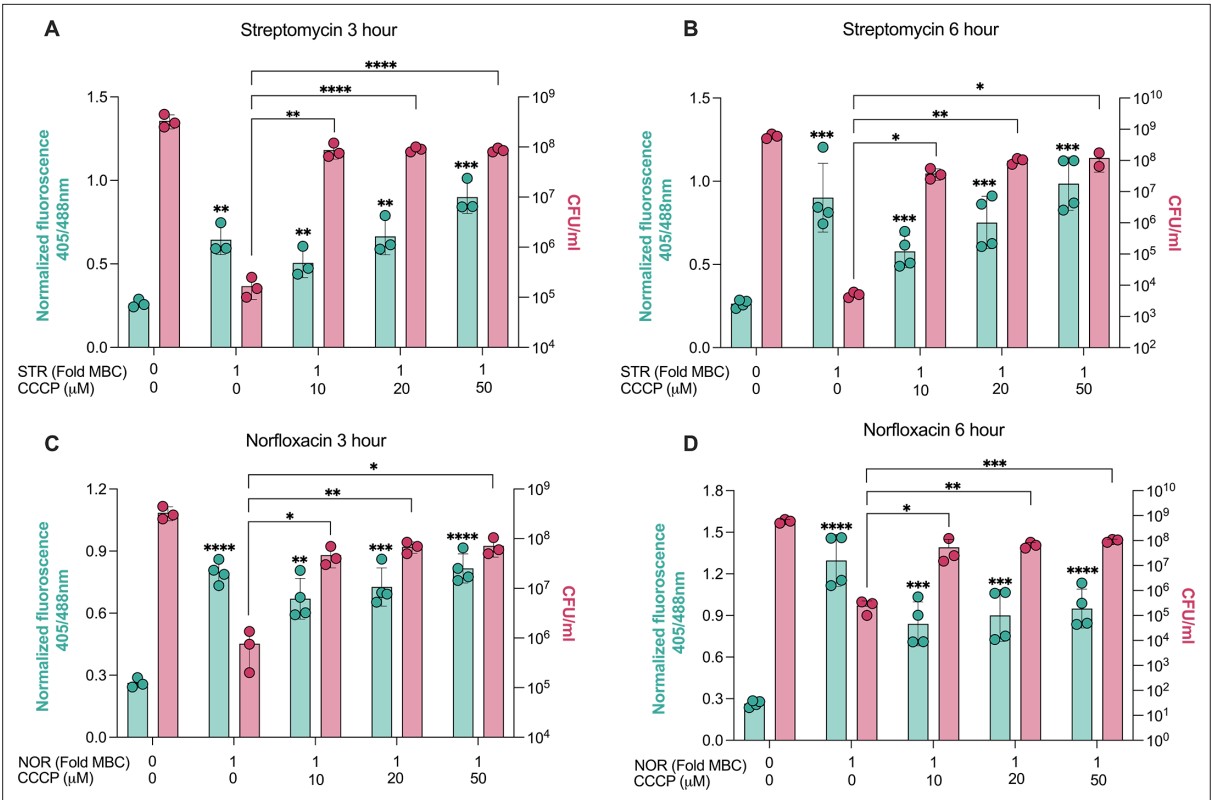

**Figure 7.** Norfloxacin and streptomycin-induced reactive oxygen species (ROS) is insufficient to mediate cell death. Ratiometric response of *M. smegmatis* expressing Mrx1-roGFP2 redox biosensor (left y-axis) and the cell viability (right y-axis) in response to streptomycin (STR) (**A, B**) and norfloxacin (NOR) (**C, D**) with and without the co-supplementation with carbonyl cyanide 3-chlorophenylhydrazone (CCCP) post 3 hr and 6 hr of treatments. Data points represent the mean of at least three independent replicates ± SD. *$p<0.05$, **$p<0.01$, ***$p<0.001$, ****$p<0.0001$ by Student' t-test (unpaired).

The online version of this article includes the following figure supplement(s) for figure 7:

**Figure supplement 1.** ROS is not the dominant driver of antibiotic lethality.

## ATP burst and not ROS is the dominant driver of antibiotic lethality in mycobacteria

Antibiotic-induced ROS has been demonstrated to trigger rapid bacterial cell death, superseding the damage caused by the inhibition of vulnerable primary targets (*Kohanski et al., 2007*; *Lobritz et al., 2015*). After demonstrating that inhibiting either ROS generation or excess ATP production can inhibit antibiotic-induced cell death, we sought to determine which of these two mechanisms is the dominant driver of lethality. Distinguishing between the effects of ROS and ATP becomes challenging due to their common origin from aerobic respiration. For this, ROS levels were compared under the conditions of ATP burst formation (antibiotic treatment alone) and ATP burst mitigation (antibiotic plus CCCP/nigericin or in *ΔatpD M. smegmatis*). We reasoned that if damage by ROS is the dominant mechanism driving cell death, then antibiotic treatment in *ΔatpD M. smegmatis* or co-treatment with CCCP/nigericin would decrease ROS levels to impart enhanced survival in the presence of antibiotics. Interestingly, while the antibiotic treatment alone increased fluorescence ratio at 405/488 nm and caused cell death (*Figure 7A–D*), CCCP-protected cells also had elevated 405/488 nm ratio (oxidative stress), indicating that the ATP burst, and not ROS production, is the dominant mechanism driving antibiotic lethality. Similar to CCCP, increased ratios of fluorescence at 405/488 nm were obtained with antibiotic and nigericin co-treatment (*Figure 7—figure supplement 1A*), as well as in the *ΔatpD M. smegmatis* exposed to antibiotics (*Figure 7—figure supplement 1B*). In all cases, antibiotic-induced ATP burst was minimised; however, the ROS levels remained higher in the surviving cells. To reciprocally test that ROS is not the dominant driver of antibiotic action, we examined the effects of co-treatment with 10 mM GSH on antibiotic-induced ATP bursts. The antioxidant GSH not only

reduced oxidative stress but also prevented the ATP burst caused by antibiotic treatment (*Figure 7—figure supplement 1C*). Thus, GSH-mediated rescue of cell survival upon antibiotic treatment could be attributed to both the reduction of oxidative stress as well as the inhibition of the ATP burst, and therefore, antibiotic-induced ATP burst is the dominant driver of antibiotic action.

Together, these results suggest that oxidative stress alone, a correlate of ROS levels is insufficient to cause cell death on its own and is not the sole mediator of antibiotic action. It points out that ROS is a by-product of antibiotic action (i.e. enhanced respiration) rather than the predominant secondary mediator of cell death, as originally understood. Notably, the CCCP-mediated increase in respiration was also evident from a dose-dependent increase in 405/488 nm ratios with CCCP treatment alone (*Figure 7—figure supplement 1D and E*). Importantly, our results do not rule out the generation and deleterious effects of ROS upon antibiotic exposure but rather suggests to emphasise that antibiotic lethality is dominantly mediated by excessive ATP levels, and in the absence of it, mycobacteria can survive antibiotic-induced ROS for longer periods of time. In summary, our results show that NOR and STR treatments increase respiration, resulting in a lethal ATP burst that drives mycobacterial death.

## Antibiotic-induced ATP burst deprives cells of essential divalent metal ions

ATP synthesis is essential for optimal growth and survival against stresses in all living organisms. Therefore, the involvement of ATP in cell death is difficult to comprehend. However, ATP is known to bind and chelate various divalent metal ions (*Jahngen and Rossomando, 1983*; *Distefano and Neuman, 1953*; *Khan and Martell, 1962*) and intriguingly, a study reported growth inhibitory activity of exogenously supplied ATP on multiple bacterial species, which was rescued upon co-treatment with magnesium and iron salts (*Tatano et al., 2015*). Based on these studies, we hypothesised that the ATP surge observed upon antibiotic treatment may have sequestered and deprived the macro-molecules and cellular processes of their essential divalent metal ions, resulting in cell death. Towards this end, we first established whether metal ion chelation alone is lethal to cell survival. Since the metal ion chelators, EDTA and EGTA are not taken up by Mycobacteria, as a proof-of-concept, we employed the cell-permeable iron chelator, bipyridyl (BP), to test whether intracellular iron depletion causes bacteriostasis or cidality. The *M. smegmatis* strain challenged with increasing concentrations of bipyridyl displayed a dose-dependent killing (*Figure 8A*), demonstrating that iron removal causes cell death. Next, we reasoned that if ATP burst-mediated cell death was due to metal ion depletion, then exogenous supplementation of divalent metal ions should increase bacterial survival to antibiotics. To test this, we compared cell survival upon NOR or STR treatment with and without the co-supplementation of four different divalent metal ions: $Fe^{2+}$, $Mg^{2+}$, $Mn^{2+}$, and $Zn^{2+}$. We observed a concentration-dependent protection from antibiotics provided by all four metal ions, suggesting that elevated ATP levels contribute to cell death in part by sequestering divalent metal ions. (*Figure 8D and E*). A similar level of protection was not achieved from monovalent metal ions (*Figure 8—figure supplement 1*). Separately, we also tested whether divalent metal ion supplementation was toxic to cells and caused growth inhibition, leading to increased cell survival. Notably, the high salt concentration of 5 mM used in our experiment had no adverse effect on the bacterial growth rate (*Figure 8F*), and thus, the enhanced survival obtained with salt co-treatment was not due to any growth arrest or metabolic inhibition by metal ions. A decrease in cell survival observed with the co-treatment of 10 mM $Fe^{2+}$ could be attributed to the antibiotic-induced Fenton cycle-mediated ROS generation resulting from excessive free iron in the cells. This observation further suggests that co-treatment with up to 10 mM salts did not hinder antibiotic uptake, and the enhanced survival observed with metal ions was unlikely to be due to reduced drug uptake. We should also note that the elevated ATP levels may affect multiple other physiological processes in bacteria; for instance, the status of protein phosphorylation or allosteric activity of metabolic enzymes. However, studying the impact of ATP burst on different aspects of mycobacterial physiology upon antibiotic exposure is beyond the scope of this study and demands a more focused investigation.

## $^{13}$C isotopomer analysis suggests increased metabolic flexibility and bifurcation of TCA cycle flux as a bacterial adaptive mechanism

After studying the nature of antibiotic-induced stresses, we sought to investigate bacterial counter-responses that could promote adaptation to antibiotics and growth recovery. $^{13}$C isotopologue profiling

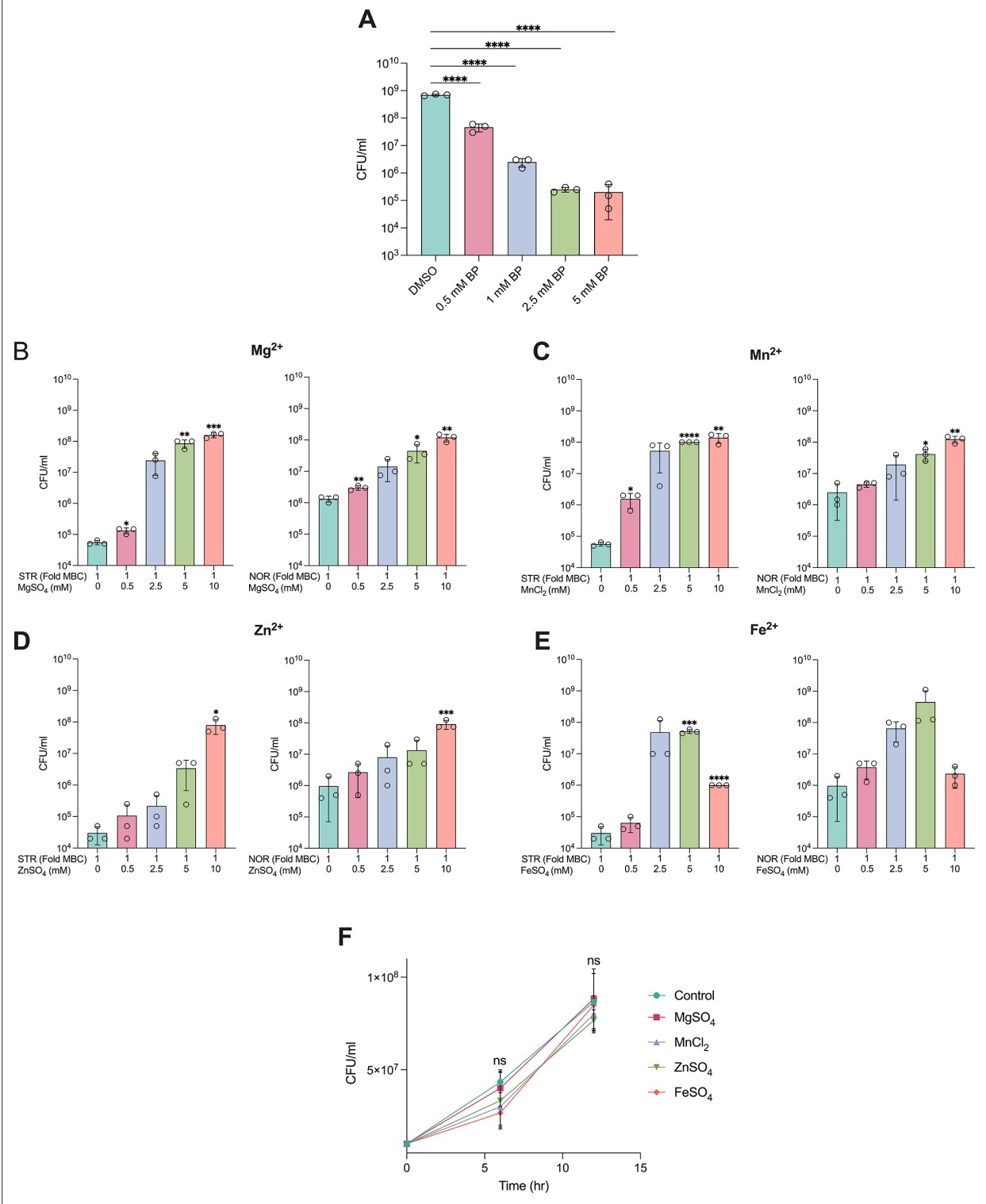

**Figure 8.** Antibiotic-induced ATP levels chelate divalent metal ions. (**A**) Effect of increasing concentrations of bipyridyl (BP) for 6 hr on the survival of *M. smegmatis* (0.8 OD/ml). Bar graphs indicate the effect of increasing concentrations of $MgSO_4$ (**B**), $MnCl_2$ (**C**), $ZnSO_4$ (**D**), and $FeSO_4$ (**E**) on the survival of *M. smegmatis* (0.8 OD/ml) challenged with lethal doses of the antibiotics for 6 hr. Growth curve (**F**) showing the effect of 5 mM of $FeSO_4$, $MgSO_4$, $MnCl_2$, $ZnSO_4$ on the survival of *M. smegmatis*. Data points represent the mean of at least three independent replicates ± SD. Statistical significance was calculated using Student' t-test (unpaired). *$p<0.05$, **$p<0.01$, ***$p<0.001$, ****$p<0.0001$.

The online version of this article includes the following figure supplement(s) for figure 8:

*Figure 8 continued on next page*

*Figure 8 continued*

**Figure supplement 1.** Effect of 5 mM potassium chloride on the survival of *M. smegmatis* challenged with 1 X MBC$_{99}$ of streptomycin (STR) and norfloxacin (NOR) for 6 hr.

was performed to investigate the metabolic rewiring that promotes the resumption of growth during the recovery phase. Using $^{13}$C glucose as a carbon tracer, intracellular metabolites from central carbon metabolism (at t=25 hr, treated with ¼x MBC$_{99}$ of NOR or STR) were assessed. In agreement with the proteomics data, there was an increase in citrate flux (*Figure 9A*) that correlated with the upregulation of citrate synthase (*Figure 3F*). Similarly, there was an increase in flux through the oxidative segment of the TCA cycle and towards alpha-ketoglutarate/glutamate production, which correlated with the observed increases in aconitase and isocitrate dehydrogenase levels. The time-dependent increase in the ratio of NADH/NAD+ under NOR and STR treatments (*Figure 9B*) and a dose-dependent increase with STR treatment (*Figure 9C*) further confirmed the increased flux measurements through the TCA cycle. The observed increase in the levels of NADH, as also confirmed by mycobacteria-specific in vivo NADH biosensor, Peredox (*Figure 9—figure supplement 1A*; *Bhat et al., 2016*), can fuel the ETC chain further to enhance the production of ATP and generate ROS as a by-product. Collectively, our measurements of increased respiration, ROS generation, and ATP production could be explained by an increase in flux through the TCA cycle, following sub-lethal antibiotic treatment.

The decreased flux into pyruvate indicates that phosphoenolpyruvate carboxykinase (PckA), which was upregulated in the proteomic analysis (*Figure 3F*), is operating in the anaplerotic direction to fix CO$_2$ to form oxaloacetate (OAA). It has been demonstrated that the redox state of the cells influences the reversal of PEPCK activity and has been associated with increased ROS production (*Machová et al., 2014*). This finding was further corroborated by the increase in the aspartate label (a surrogate for oxaloacetate) and the M+3 and M+5 ionic species of $^{13}$C-citrate (*Figure 9—figure supplement 1B and C*). This increased flux into OAA can facilitate greater utilisation of acetyl-CoA by the TCA cycle, as indicated by the increased label of citrate. Since antibiotic-mediated death was caused by increased respiration, ROS, and ATP levels, we expected that bacteria exposed to sub-lethal doses of antibiotics may remodel their central carbon metabolism (CCM) to minimize respiration and improve their chances of survival. In agreement with this hypothesis, the enhanced flux in the oxidative arm of the TCA cycle (alpha-ketoglutarate) was found to diverge towards glutamate, glutamine, and GABA shunt following streptomycin treatment (*Figure 9A*). This deviation of the flux was facilitated by upregulation of glutamate dehydrogenase, glutamate synthase, glutamine synthetase, and glutamate decarboxylase, as evident from the proteomics data (*Figure 3F and G*). The amino acids glutamate and glutamine direct the carbon flux to amino acid biosynthesis and help regulate energy and nitrogen metabolism (*Borah Slater et al., 2023*; *Viljoen et al., 2013*; *Gallant et al., 2016*). In addition, increased flux in glutamate and GABA would help bypass two enzymatic reactions of the TCA cycle's oxidative arm, restricting ROS production and regulating anaplerosis (*Cao et al., 2013*; *Ramond et al., 2014*). Moreover, we observed an increased flexibility of the TCA cycle in switching carbon flux from aspartate/oxaloacetate to alpha-ketoglutarate/glutamate in response to antibiotics. This reaction is catalysed by aspartate aminotransferase (AspAT), which keeps balance between the cataplerosis and anaplerosis of TCA cycle intermediates in mycobacteria (*Jansen et al., 2020*). AspAT was also found to be upregulated in our proteomics analysis (*Figure 3F*). Overall, our results suggest that by enhancing expression and activity of TCA cycle enzymes and those of the anaplerotic node (*Basu et al., 2018*), the antibiotic-adapted central carbon metabolism shows more plasticity and resilience in terms of minimizing the production of ROS and ATP, while still maintaining the overall metabolism and growth.

The glyoxylate shunt and PPP pathway were found to be involved in antibiotic tolerance mechanisms (*Lee et al., 2019b*; *Nandakumar et al., 2014*; *Mackenzie et al., 2020*). We observed a decrease in the incorporation of $^{13}$C label into succinate, indicating a decreased flux through succinate-semialdehyde dehydrogenase (SSADH), alpha-ketoglutarate dehydrogenase (KGDH), or isocitrate lyase (ICL1). Antibiotic treatments decreased the expression of the SucA and SucB subunits of KGDH and the GabD1 and GabD2 subunits of SSADH (*Figure 3F*), explaining the lower flux in succinate. Furthermore, there were no changes in the $^{13}$C label in histidine and sedoheptulose 7 P indicating that flux through the PPP remained unperturbed upon antibiotic treatment (*Figure 9—figure supplement 1D*). Since the enzymes of the oxidative stage (G6-PDH and 6-PGD) of the PPP were found upregulated in our

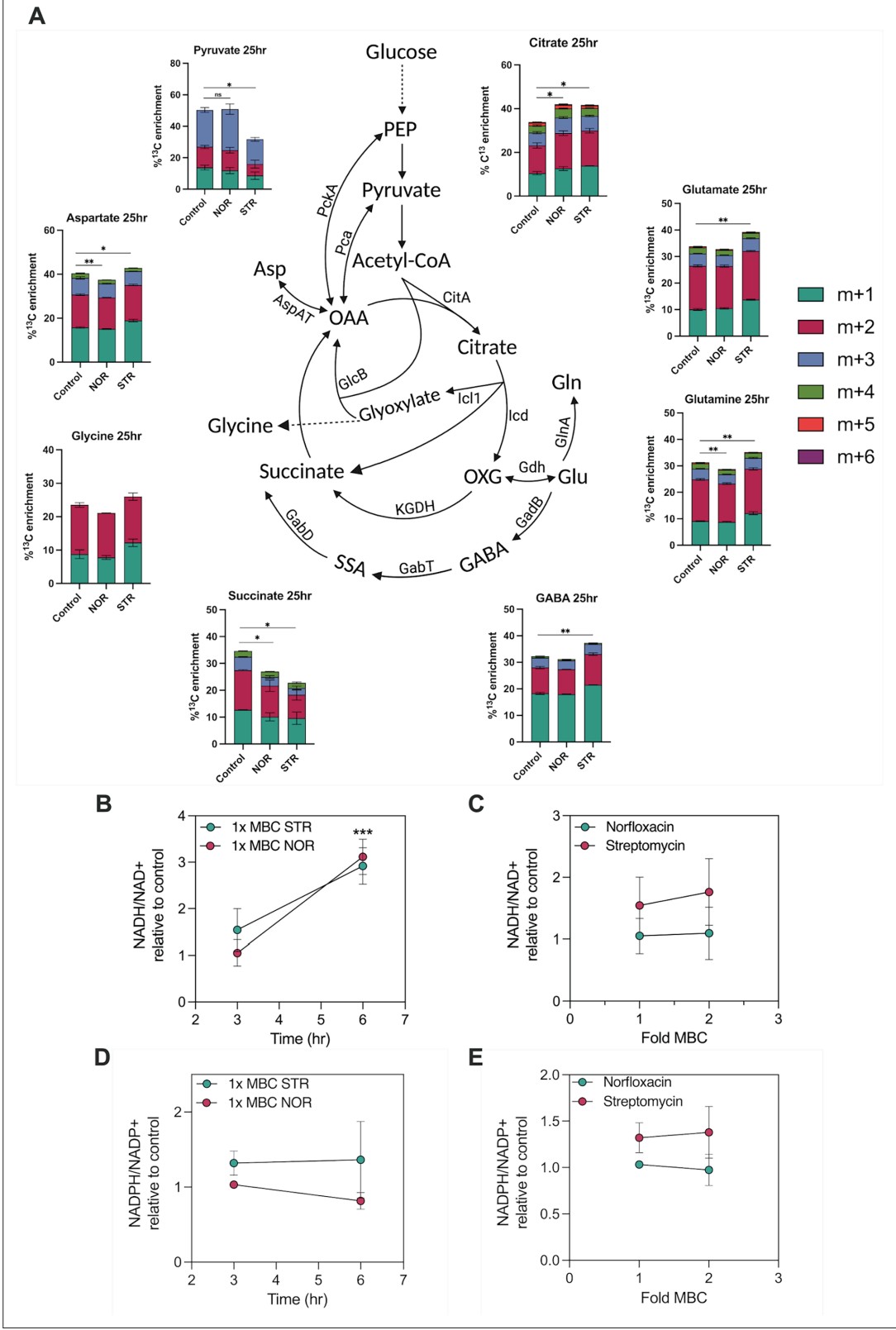

**Figure 9.** ¹³C metabolomics suggests increased metabolic flexibility and bifurcation of TCA cycle flux as bacterial adaptive mechanism. (**A**) Percent ¹³C enrichment in central carbon metabolism (CCM) metabolites of *M. smegmatis* challenged with ¼x MBC₉₉ of antibiotics at 25 hr time point (recovery phase). 'M' denotes the molecular mass of the metabolites, where M+1(n) denotes the incorporation of ¹³C labelled carbon in the metabolites, leading to

*Figure 9 continued on next page*

*Figure 9 continued*

increase in the mass by that extent. Data for each isotopologue are represented as mean ± SD from independent triplicates. PEP, phosphoenolpyruvate carboxykinase; OXG, alpha-ketoglutarate; Glu, glutamate, Gln, glutamine, OAA, oxaloacetate; Asp, aspartate; GABA, γ-Aminobutyric acid, SSA, succinic semialdehyde. Line graphs represent the ratios of NADH/NAD+ and NADPH/NADP+ with time (**B, D**) and dose-dependent manner (**C, E**), respectively, after antibiotic treatments. Data points represent the mean of at least three independent replicates ± SD. *$p<0.05$, **$p<0.01$, ***$p<0.001$ were calculated by Student's t-test (unpaired).

The online version of this article includes the following source data and figure supplement(s) for figure 9:

**Source data 1.** Normalised abundance of metabolites identified.

**Figure supplement 1.** Antibiotic-induced metabolic changes in the central carbon metabolism of *M. smegmatis.*

proteomics analysis, it may be speculated that the flux from ribose 5-phosphate got channelised towards purine biosynthesis, which is consistent with the enrichment of the purine metabolic pathway in response to STR (*Figure 3—figure supplement 1B*). The NADPH/NADP+ ratio measurements also revealed no significant differences in antibiotic-treated cells (*Figure 9D and E*). Overall, no changes in flux into the PPP and glyoxylate shunt were observed during the antibiotic exposure. A complete list of identified metabolites and $^{13}C$ enrichment of each isotopologue is available as *Figure 9—source data 1*.

## Sub-lethal antibiotic exposure can potentiate development of tolerance and genetic resistance to antibiotics

Next, we questioned whether sublethal antibiotic concentrations have activated any intrinsic drug resistance mechanisms. Specifically, proteomics data analysis revealed a 50-fold upregulation of the protein Eis at 25 hr after STR treatment (*Figure 10A*). Eis is known to acetylate and inactivate aminoglycosides, thus conferring antibiotic tolerance and promoting enhanced intracellular survival (*Schrader et al., 2021*; *Zaunbrecher et al., 2009*). This provides evidence that mycobacteria may induce intrinsic mechanisms to adapt to sub-lethal antibiotic exposure.

To test whether sub-lethal antibiotic concentrations increase mutation frequency, we measured mutation frequency using rifampicin resistance as the phenotypic reporter. We harvested cells from the recovery phase (25th hr) and compared the number of rifampicin-resistant mutants in NOR/STR-treated versus untreated conditions. Within 22.5 hr of antibiotic exposure, the mutation frequency of cells adapted to sub-lethal concentrations of STR increased by fourfold, whereas NOR treatment did not significantly increase the mutational frequency (*Figure 10B*). These findings indicate that antibiotic-induced physiological adaptation may facilitate the acquisition of drug-specific qualitative mutations or non-specific quantitative mutations in metabolic enzymes, leading to antibiotic resistance and treatment failure.

## Discussion

The current understanding of antibiotic lethality is limited to target corruption as the primary cause and free radical-induced cellular damage as a secondary consequence. This study, while exploring the mycobacterial responses to antibiotics, identifies ATP burst as the new and dominant driver of antibiotic-induced cell death in mycobacteria. Our differential proteomics analysis upon antibiotic exposure revealed a rewiring in the central carbon metabolism and generation of an anti-oxidant response due to the production of ROS. Damage by ROS has been proposed as an universal mechanism that many bactericidal antibiotics use to eliminate bacteria, irrespective of their primary targets (*Kohanski et al., 2007*). Respiration-inducing molecules, such as Vitamin C, N-acetylcysteine (NAC), and other thiols were found to use ROS to enhance antibiotic lethality, even against drug-induced persisters, while ROS quenching by antioxidants rescued bacteria from cell death (*Shee et al., 2022*; *Vilchèze et al., 2013*; *Nandakumar et al., 2014*; *Vilchèze et al., 2017*; *Grant et al., 2012*). Our measurements revealed upregulated central carbon metabolism enzymes, increased flux through the TCA cycle, high NADH/NAD +ratios, and elevated ROS levels in response to antibiotic. Notably, the set of bacterial responses were shared by norfloxacin and streptomycin, which belong to two distinct classes of drugs, the fluoroquinolones and aminoglycosides, respectively. Thus, our observations further strengthen the evidence of increasing involvement of ROS in antibiotic action and also in

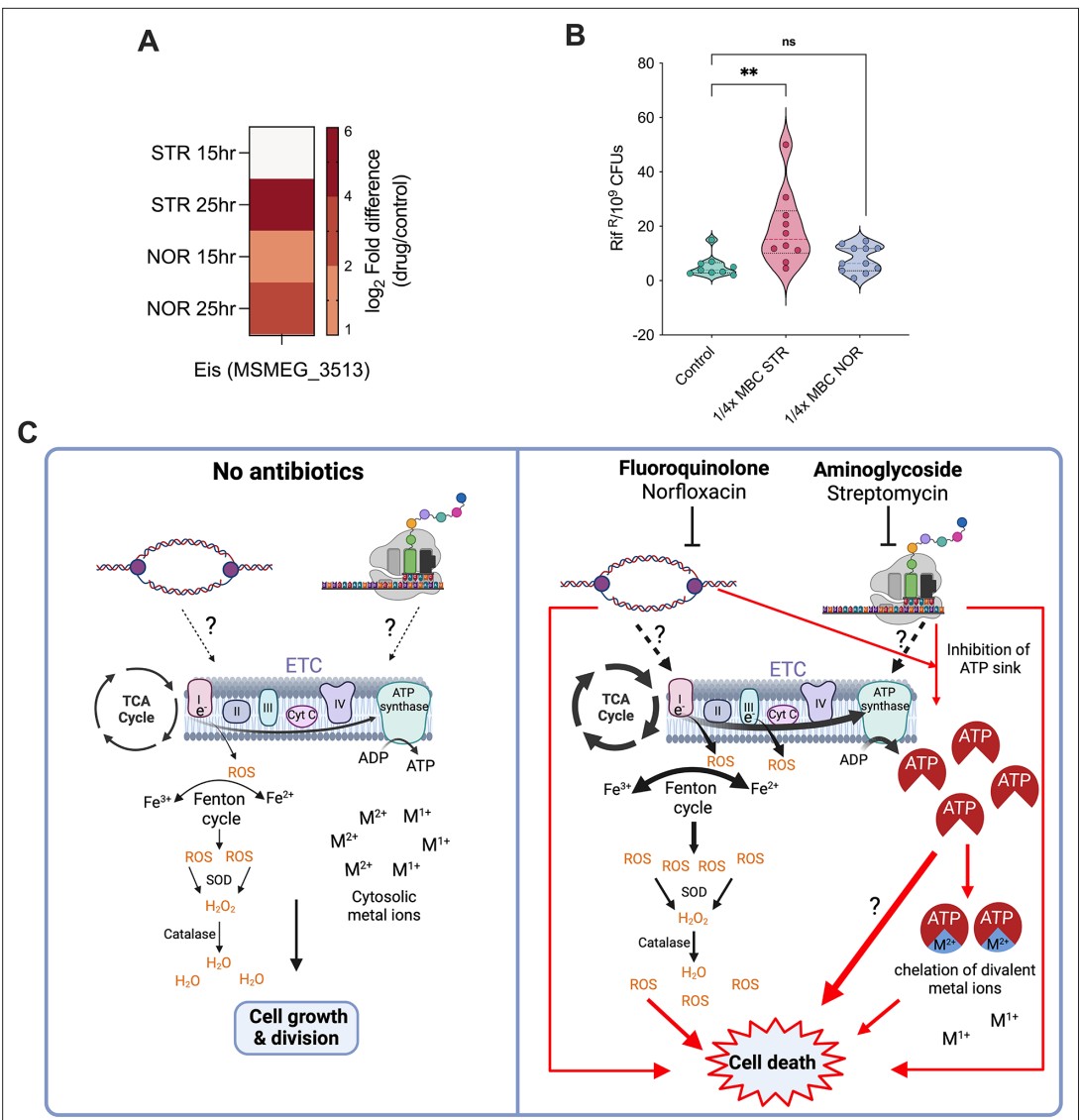

**Figure 10.** Sub-lethal antibiotic exposure can potentiate the development of antibiotic resistance. Heat map (**A**) showing the $\log_2$ fold differences of protein Eis in response to antibiotic treatment. Panel (**B**) depicts the mutational frequency of *M. smegmatis* grown in ¼ x $MBC_{99}$ of the antibiotics for 25 hours; each dot represents the mutational frequency of one replicate (n=10). *$p<0.05$, **$p<0.01$, ***$p<0.001$ were calculated by Student's t-test (unpaired). (**C**) Proposed extended model of antibiotic lethality in Mycobacteria. Panel on left explains the interaction among the central dogma processes and central carbon metabolism (CCM), where, depending on the need for energy, flux through CCM is modulated leading to optimal growth and division. Panel on right explains the effect of inhibition of essential cellular processes by antibiotics, resulting in activation of CCM and respiration, leading to reactive oxygen species (ROS) generation and ATP burst. An increase in ATP levels is also contributed by the inhibition of ATP-consuming central dogma and cell division processes by antibiotics. Elevated ATP levels in part chelate essential divalent metal ions and can deny them as co-factors for various proteins, eventually leading to cell death. The thickness of arrows indicates the impact of the processes.

inducing tolerance in bacterial pathogens, including mycobacteria (*Shee et al., 2022*; *Nandakumar et al., 2014*; *Rowe et al., 2020*; *Mishra et al., 2019*).

What is the trigger for high respiration and ROS production? While electron leakage from the ETC is the primary source of ROS, it is unclear how antibiotics are sensed beyond their primary targets and how they modulate the ETC to generate ROS. Recent studies have demonstrated that antibiotic treatments deplete the purine nucleotide pool, resulting in an increase in purine biosynthesis and hence ATP demand (*Lobritz et al., 2022*; *Yang et al., 2019*). This increased energy demand becomes a significant reason for increased respiration, oxygen consumption, and ROS generation, thus contributing to enhanced antibiotic lethality (*Lobritz et al., 2022*). Furthermore, the increased purine biosynthesis may increase levels of AMP, reducing the ATP/AMP ratio, which could stimulate

the rate-limiting steps of glycolysis and TCA cycle to enhance aerobic respiration, ROS production, and ATP synthesis. Similarly, a down-regulation of proteins involved in central dogma and cell division processes in response to sub-lethal levels of NOR or STR could be interpreted as an indication of diminished availability of biosynthetic building blocks and energy and may have triggered increased metabolic activity as a compensatory mechanism. Therefore, perturbing the target function by antibiotics could lead to multiple stimulating impacts on bacterial metabolism and respiration, originally intended as a bacterial response, resulting in its fatality.

To confirm that increased respiration led to ROS production, we directly measured ATP levels, the end product of aerobic respiration. Both norfloxacin and streptomycin treatments resulted in a substantial dose- and time-dependent rise in ATP levels that was inversely proportional to cell survival. By inhibiting antibiotic-induced ATP burst using a protonophore uncoupler of ETC and oxidative phosphorylation (CCCP and nigericin), an inhibitor of ATP synthesis (Bedaquiline), or the ATP-deficient ΔatpD strain (genetic uncoupler), we showed that the observed increase in ATP concentration was due to enhanced oxidative phosphorylation and significantly contributed to antibiotic action and cell death.

Growth rate and metabolism were shown to influence antibiotic efficacy (*Greulich et al., 2015*; *Lee et al., 2018*; *Haugan et al., 2019*). A previous study concluded that the bacterial metabolic state, not the growth rate, is the primary driver of cidality (*Lopatkin et al., 2019*); however, the precise metabolic reactions or factors governing antibiotic lethality remain unclear. In addition to increased respiration, our proteomics analysis revealed that downregulation of ATP-consuming processes, such as the central dogma and cell division could be contributing to ATP accumulation. Thus, energy metabolism and particularly ATP synthesis become a central component of the connection between metabolic state and antibiotic susceptibility. Recent studies have observed a remarkable correlation between lower ATP levels and generation of persister population in *S. aureus* and *E. coli* (*Conlon et al., 2016*; *Shan et al., 2017*). Interestingly, exogenous supplementation of alanine or glucose increased the susceptibility of drug-resistant *E. tarda* by boosting the flux through the TCA cycle and increasing the proton motive force, resulting in increased kanamycin uptake (*Peng et al., 2015*). Based on these observations, it may be logical to reason that inhibition of oxidative phosphorylation would result in reduced antibiotic cidality and reveal a plausible direct role of ATP in determining antibiotic susceptibility. Supporting this hypothesis, cell wall-acting antimycobacterials were shown to increase ATP levels in *M. bovis BCG*, which directly correlated with antibiotic efficacy (*Shetty and Dick, 2018*). Furthermore, energy metabolism inhibitors, bedaquiline and telacebec were found to dampen the bactericidal activity of isoniazid and moxifloxacin against *M. bovis BCG* and *M. tuberculosis* (*Lee et al., 2019a*). These studies, however, fell short in establishing a time and dose-dependent relationship between antibiotics and ATP levels, and also in adequately distinguishing the specific impact and significance of ATP levels in antibiotic lethality compared to overall metabolic and respiratory activities, as well as levels of ROS.

Since inhibition of either ROS generation or ATP burst was sufficient to reduce the lethality of norfloxacin and streptomycin, we sought to determine which of the two is the dominant contributor. Intriguingly, measurements of ROS in CCCP-protected mycobacteria revealed that elevated ROS levels were insufficient to cause cell death upon antibiotic treatment in the absence of an ATP burst, indicating that ATP burst is the dominant mechanism driving cell death. Conditions, such as hypoxia, stationary phase, nutrient starvation, biofilm formation (*Lee et al., 2019b*; *Ojha et al., 2008*; *Samuels et al., 2022*) or an artificial reduction of the NADH/NAD+ ratio (*Shee et al., 2022*) render mycobacteria less susceptible to antibiotics compared to actively growing cells, and this phenotype has been attributed to lower levels of ROS. Based on our findings, we speculate that these conditions would also generate lesser flux through the electron transport chain (respiration), which, in addition to decreasing ROS levels, would also produce lower levels of ATP. Since the metabolic rate, ROS, and ATP burst are interrelated, studying their individual contributions remains challenging but necessary. We show that in the absence of an ATP burst, mycobacteria can tolerate the consequences of primary target inhibition and the damage from ROS for a longer duration, making them insufficient to cause rapid cell death. Contrary to our observations, a genetic uncoupling of ATP synthesis potentiated lethality in *E. coli*, indicating that enhanced respiration and ROS production is the dominant driver of efficacy for bactericidal antibiotics in *E. coli* (*Lobritz et al., 2015*). These findings suggest similarities in the bacterial metabolic toolkit exploited by cidal antibiotics while targeting across genera, yet

identify the differences in the pivotal component that drives cell death in *M. smegmatis*. Therefore, the secondary mechanisms of lethality should be tested for a panel of bactericidal antibiotics, with distinct mode of action, across different bacterial pathogens, including *M. tuberculosis*, which is an intriguing lacuna of our study. Likewise, while we note the commonality in the involvement of ROS and ATP burst in fluoroquinolones and aminoglycoside action, we highlight that the extent of ROS and ATP's contribution may differ between the two antibiotics. The levels of ROS produced were higher for norfloxacin, whereas the extent of ATP burst was greater for streptomycin, which demanded additional bedaquiline to suppress the streptomycin-induced ATP burst and killing.

Metal ion binding by ATP is a well-known phenomenon, where ATP forms a stable coordination complex with the divalent metal ion (*Distefano and Neuman, 1953*; *Khan and Martell, 1962*; *Nanninga, 1957*). We observed that co-supplementation with divalent metal ions rescued mycobacteria from the cidality of STR and NOR. Further, exogenous supplementation with ATP was found to cause bacteriostasis in various bacteria, including mycobacteria (*Tatano et al., 2015*). Based on this observation, we propose metal ion chelation and depletion by ATP burst as a novel bactericidal mechanism executed by antibiotics in mycobacteria. We have separately established that the depletion of $Fe^{2+}$ is lethal in mycobacteria, suggesting the essentiality of these cofactors for cell survival. Future studies should investigate the comprehensive mechanism of antimicrobial action of excessive ATP. For instance, a recent study found increased ATP levels generated upon aminoglycoside treatment resulting in ATP hydrolysis through the reversal of $F_0F_1$-ATP synthase activity, causing hyperpolarisation of the membrane and cell death in *E. coli* (*Bruni and Kralj, 2020*).

In conclusion, our study fills a gap and expands the current understanding of how antibiotics ultimately cause cell death in the context of *M. smegmatis*. Whether other bactericidal antibiotics employ ATP burst to mediate their lethality in pathogenic mycobacteria and other serious pathogens warrants further investigation. In conjunction, compounds that synthetically generate ATP burst should also be tested for antimycobacterial efficacy.

*M. tuberculosis* is an outlier among multi-drug resistant pathogens due to its low recombination rate, limited genetic diversity, and absence of an accessory genome. Becoming a successful pathogen, despite its inability to rapidly evolve and withstand new antibiotics, alludes to the presence of an effective innate mechanism of resistance. Low fitness-cost mutational events causing transient antibiotic tolerance in the absence of a strong selective pressure precede and possibly boost successive qualitative mutations on antibiotic targets (*Kohanski et al., 2010*; *Levin-Reisman et al., 2017*). In our observations, Mycobacterial counter responses included downregulation of central dogma and cell division processes, thereby reducing the vulnerability of the target and thus efficacy of the antibiotic. Reduction of the energy-consuming processes could reduce the anabolic demand and help decrease their energy metabolism, ROS production, and ATP synthesis. $^{13}$C isotopomer analysis revealed rerouting of flux from alpha-ketoglutarate to glutamate-glutamine and GABA, bypassing the oxidative arm of the TCA cycle. We also measured an increased mutation frequency and the induction of Eis, demonstrating that the antibiotic-adapted population may facilitate the selection of drug-resistant mutants (*Kohanski et al., 2010*; *Levin-Reisman et al., 2017*; *Schrader et al., 2021*; *Zaunbrecher et al., 2009*). In summary, we observed that upon exposure to sub-inhibitory levels of antibiotics, Mycobacterium transiently switched to a quiescent state that could reduce its respiration, thus denying the antimicrobial effect of ROS and metal ion chelation by ATP. These findings could be used to develop antibiotic adjuvants that enhance aerobic metabolism (*Kitzenberg et al., 2022*) and increase ROS-mediated damage and ATP production. Our study also advocates for a thorough understanding of the proximate and ultimate causes of antibiotic lethality (*Baquero and Levin, 2021*), in order to avoid drug combination with overlapping (secondary) actions and mechanism of tolerance.

# Materials and methods
## Bacterial strains, growth conditions

*Mycobacterium smegmatis* $MC^2155$ (Msm) strain was grown in Middlebrook 7H9 broth (Difco) supplemented with 2% glucose, 0.05% Tween-80 at 37 degree incubator at 200 rpm shaking or on Middlebrook 7H10 agar supplemented with 2% glucose. Msm cells were electroporated with Mrx1-roGFP2 (Hyg$^R$) or Peredox-mCherry biosensor and grown with 50 µg/ml of hygromycin. Msm cells were electroporated with PHR-mCherry (ATP/ADP sensor) plasmids and grown with 20 µg/ml of kanamycin.

Wild-type Msm and Δ*atpD* Msm (Hyg$^R$) were electroporated with Mrx1-roGFP2 (Kan$^R$) grown with 20 µg/ml of kanamycin. All three biosensors constitutively express their respective sensor proteins under the *hsp60* promoter and offer ratiometric measurements.

## Determination of MIC$_{99}$ and MBC$_{99}$

A secondary culture of Msm was grown till the exponential phase (0.6 OD$_{600nm}$ /ml). MIC$_{99}$ was determined by incubating Msm cells (OD$_{600nm}$ = 0.0005 /ml) with twofold serially diluted antibiotic concentrations for 36 hr. Percentage survival was measured by REMA assay. The drug concentration that inhibited 99% of bacterial growth was termed as MIC$_{99}$. For the determination of MBC$_{99}$, Msm cells from the exponential phase were challenged with various drug concentrations and incubated at 37 degrees in a 96-well plate for 24 hr. The drug concentration that killed 99% (2log$_{10}$ fold killing) of the initial bacterial population was termed MBC$_{99}$. For the experiments requiring different initial inoculum, a corresponding MBC$_{99}$ was determined to account for inoculum effect (that can influence the extent or duration of killing).

## Growth curve of *M. smegmatis* with sub-lethal concentrations of norfloxacin and streptomycin

For the growth curve experiment, a tertiary culture of Msm cells was exposed to either 1x-MBC$_{99}$ or ½x-MBC$_{99}$ or ¼x-MBC$_{99}$ concentrations of norfloxacin and streptomycin, individually, with starting inoculum of 0.0025 /ml. Cultures were grown at continuous shaking conditions at 37 °C. Drugs were added at 2.5 hr after the start of the growth curve. For the measurement of viability, cells were serially diluted and plated on LB agar at every 2.5 hr for a total of 25 hr. The experiment was repeated at least six times for both antibiotics.

## MIC verification

MIC verification was performed and compared between cells grown with and without the ¼x MBC$_{99}$ concentrations of norfloxacin and streptomycin, individually, for a total of 25 hr. Cells were harvested from the 25 hr time point and seeded into 96-well plates, pre-seeded with twofold serially diluted antibiotic concentrations. Percentage survival was plotted using the REMA assay.

## Quantitative proteomics

### Reagents

All reagents were purchased from Sigma Merck. Sequencing grade modified trypsin was purchased from Promega. All the MS grade solvents, water, acetonitrile, and formic acid were procured from Fisher Chemicals.

### Sample preparation

Proteomic analysis was performed as reported previously (*George et al., 2022*). For proteomics, cells were grown in five independent experiments each for drug-treated and untreated conditions. Msm cells were harvested from desired time points and conditions at 6000 g at 4 °C, washed thrice with chilled PBS (pH = 7.4). The cell pellets were stored at –80 °C until further use. Cell pellets were resuspended in 1 ml of lysis buffer (PBS + protease cocktail inhibitor + 1 mM EDTA) to minimise non-specific protein degradation. Cells were lysed by mechanical shearing in a bead-beater with at least 10 cycles (7 m/s for 45 s) of lysis. Tubes were kept on ice for 3–4 min in between the cycles. The supernatant of the cell lysate was precipitated using 1:4 v/v of TCA. Protein precipitates were solubilised in 6 M urea and subsequently quantified using BCA kit (Sigma). Ten micrograms of the protein samples were reduced with 10 mM dithiothreitol (DTT) and alkylated with 37 mM iodoacetamide (IAA). These linearised proteins in 50 mM ammonium bicarbonate (pH = 7.8), were digested with a modified MS grade trypsin (Promega; 1:5 enzyme/protein) overnight at 37 °C. The reactions were terminated by the addition of 2 µl of trifluoroacetic acid (TFA) to bring pH <2.

### Peptide desalting

Tryptic peptides were desalted using custom-made stage tips employing the Empore C-18 disks. Stage tips were conditioned and equilibrated with 100% acetonitrile and 0.1% formic acid in water,

respectively, before sample loading. Peptides were washed with 0.1% formic acid, eluted using 50% acetonitrile in 0.1% formic acid. For LC-MS analysis, the samples were dried in a vacuum centrifuge to remove acetonitrile and resuspended in 0.1% formic acid.

## LC-MS analysis

Peptides were analysed through an Ultimate 3000 RSLC nano-UPLC system connected with an Orbitrap Elite hybrid mass spectrometer (Thermo Fisher Scientific). Roughly 600 ng of peptides were loaded with automated injections onto a trap column (Acclaim PepMap 100, 3 µm particle size, 75 µm×2 cm) for 4 min, at a flow rate of 5 µl min$^{-1}$ with 0.1% formic acid. The peptide separation was achieved on a C-18 analytical column (PepMap RSLC, 2 µm particle size, 100 Å pore size, 75 µm×50 cm) at 250 nL min$^{-1}$ flow rate, using the following solvents: solvent A, 0.1% formic acid in water; solvent B, 0.1% formic acid in 100% ACN. Both columns were equilibrated for 10 min with 5% solvent B. A slow gradient from 5% to 22% of solvent B was achieved in 85 min, which was then raised to 28% of solvent B in an additional 20 min. This was followed by a quick increase to achieve 80% of solvent B in 5 min and maintained there for additional 10 min before bringing back to 2% solvent B. The analytical column temperature was maintained at 40 °C. A stable spray at a voltage of 1.8 kV was generated using the nano spray electron ionisation source (Thermo Fisher Scientific). The capillary temperature was maintained at 275 °C for effective nebulisation. The masses were measured in the positive mode using the data-dependent acquisition (DDA). MS1 and MS2 spectra were acquired in Orbitrap (60,000 resolution) and in ion trap analyser (rapid scan mode), respectively. For tandem mass spectrometry, the twenty most abundant parent ions were sequentially selected from the MS1 spectrum (350–2000 m/z) and fragmented by collision-induced dissociation (CID). For MS1, the maximum ion accumulation time was set at 100 ms with a limitation of 1x10$^6$ ions, whereas for MS2 spectra, the ion accumulation time was limited to 50 ms with a target ion limitation set at 5×10$^3$ ions. Peptide fragmentation was performed at 35% of normalised collision energy. Dynamic exclusion of 30 s was applied with a repeat count of 1 to avoid repeated analysis of the same peptide.

## Protein identification

10 raw files for each set of samples (five drug-free control and five ¼x-MBC$_{99}$ drug-treated samples) were analysed together by MaxQuant (version 2.0.3.0) using the reference proteome of *Mycobacterium smegmatis mc²155* (https://mycobrowser.epfl.ch/; version v3) through its internal peptide search engine Andromeda (*Tyanova et al., 2016a*; *Cox et al., 2014*). The following parameters were used for the protein identification: maximum missed cleavages, 2; mass tolerance for first and main search, 20 and 4.5 ppm, respectively; mass tolerance for fragment ion, 0.5 Da; variable modifications used were N-terminal acetylation and methionine oxidation; minimum unique peptide required for identification, 1; minimum peptide length 7; max. peptide mass, 4600 Da; PSM and protein identification FDR were fixed at 0.01; dependent peptide and match between runs were enabled with a match time window of 0.7 min and alignment window of 20 min.

## Relative protein quantification

The maxLFQ algorithm was enabled in MaxQuant for the label-free quantification (LFQ) of all proteins (*Lee et al., 2019b*). The protein group files generated from MaxQuant were subsequently analysed by Perseus (version: 1.6.2.2) (*Tyanova et al., 2016b*). Proteins identified as contaminants, from reverse sequence databases as well as through modified peptides were removed. LFQ intensities were log-transformed (log$_2$) and grouped in two groups (drug-free and drug-treated). Pearson correlation was performed on LFQ intensities to check for the reproducibility and correlation among samples. From five independent replicates, three highly correlating replicates were chosen for differential expression analysis using a volcano plot. Proteins with a fold difference of ±2 with *p*-value ≤0.05 were considered to be significantly dysregulated proteins.

## Exclusive expression analysis

Proteins having valid LFQ intensities in all 3 replicates of one condition and were completely absent in all the replicates of the other condition were termed as exclusively/specifically expressed in that particular condition. Similarly, proteins that were specifically repressed/absent in one condition over

another were identified. These proteins are of equal importance since they are only expressed or repressed in response to drugs.

## Enrichment analysis

Proteins that were up-regulated as well as specifically expressed in response to drugs were analysed by ClueGO, a Cytoscape plug-in, for the KEGG Pathway enrichment analysis (*Bindea et al., 2009*). Only pathways that had p-values less than 0.05 were processed further. The Benjamini-Hochberg multiple correction test was applied and the pathways that were still significant ($p \leq 0.05$) were termed as significantly enriched pathways.

## Measurement of oxidative stress (ROS)

Cells expressing Mrx1-roGFP2 biosensor were challenged with drugs and reagents at different concentrations, and fluorescence was measured at 405 nm and 488 nm excitation wavelengths for a single emission at 520 nm. The ratio of fluorescence intensities obtained at 405 nm and 488 nm (405/488 nm) for 20 mM CHP and 100 mM DTT were used to normalise the data (*Bhaskar et al., 2014*). In response to drugs or other reagents, the 405/488 nm ratio was monitored either temporally and for the endpoint in BioTek Synergy H1 Plate Reader using a black 96-well plate (Costar). Since the plate reader measurement requires a higher number of cells (100 µl of ~0.8 $OD_{600nm}$ /ml), a separate MBC determination assay was performed to determine the MBC for 0.8 $OD_{600nm}$ /ml culture density to account for inoculum effects.

## Time-kill kinetics

For the Time-kill kinetics, cells with 0.1 $OD_{600nm}$/ml (~2 × $10^7$ CFU/ml) were challenged with different antibiotic concentrations (fold MBC) alone or with other co-supplements in 96-well plates at 37 degrees for experiments involving GSH and BP. Whereas for all other experiments involving measurements of metabolic factors, such as ATP, NADH/NAD+ measurement, resazurin activity, etc, were conducted with an inoculum of 0.8 $OD_{600nm}$ and cultures were at 37 °C at 200 RPM shaking. At regular intervals, cells were aliquoted and diluted in 7H9+0.05% Tween-80 and plated on LB agar to enumerate CFUs. After 3–4 days, the colonies were counted manually and plotted as CFU/ml over the course of time to exhibit a time-kill curve.

## Measurement of ATP, NADH/NAD+, and NADPH/NADP+ ratio

Cells (0.8 $OD_{600nm}$ /ml) were challenged with various fold MBC concentrations (predetermined for 0.8 $OD_{600nm}$/ml) of norfloxacin and streptomycin. The higher inoculum was taken to ensure sufficient viability of cells upon antibiotic treatment for the detection of ATP, NAD+ and NADP+.

Intracellular ATP levels were determined using the BacTiter-Glo Microbial Cell Viability Assay (Promega) as per the manufacturer's instructions. For ATP measurement, cells from various conditions and time points were harvested and heat inactivated at 90 °C for 30 min, as performed elsewhere (*Gengenbacher et al., 2010*). 25 µl of each sample was mixed with the equal volume of a BacTiter-Glo reagent in a white flat-bottom 96-well plate (Corning). Samples were incubated in the dark at room temperature for 5 min, and bioluminescence was measured in a microplate reader (BioTek Synergy H1). The emitted luminescence was divided by the number of viable cells determined just before the heat inactivation to express ATP levels per bacterium (RLU/CFU). The CFU normalised luminescence observed for drug-free control was set as 1, and RLU/CFU values obtained for various conditions were plotted relative to the control.

NADH/NAD+ and NADPH/NADP+ ratios were measured using the Bioluminescent NAD/NADH-Glo detection kit (Promega) and NADP/NADPH-Glo detection kit (Promega), respectively, as per the manufacturer's instructions. Briefly, cells at desired conditions were lysed using two cycles of bead beating (7 m/s for 45 s). The supernatant of lysate was cleared using centrifugation and processed for the measurement of NAD+ or NADP+ and NADH or NADPH separately, and the ratio of luminescence for NADH/NAD+ and NADPH/NADP+ were plotted. The ratios obtained for controls were set as 1, and the ratios obtained for various conditions were plotted with respect to control.

## Measurement of respiratory activity using resazurin

Measurement of metabolic/respiratory activity was performed as reported elsewhere (*Schrader et al., 2021*). Briefly, cells were grown till the exponential phase in 7H9 and treated with desired drugs and supplements. At indicated time points, 100 μl aliquots of cells were removed and seeded in a black wall 96-well plate (Costar). Resazurin was added to all wells at a final concentration of 30 μg/ml and incubated in the dark at 37 °C for 30 min. The quick conversion of resazurin to resorufin was measured in a plate reader at the excitation of 530 nm and the fluorescence emission of 590 nm. Fluorescence intensities were divided by CFUs, measured just before the addition of the resazurin to express respiratory activity per bacterium. CFU normalised fluorescence intensities acquired for control were set as 1, and CFU normalised intensities obtained for various conditions were plotted relative to the control. The ability of resazurin to differentiate cells with distinct metabolic activity was separately confirmed by comparing normalised fluorescence acquired for cells from exponential phase, PBS-starved cells and heat-killed cells.

## Measurement of NADH using Peredox-mCherry biosensor

The Peredox/mCherry biosensor indicating NADH/NAD+ ratio (*Bhat et al., 2016*) was electroporated in *M. smegmatis*. Cells were grown with hygromycin (50 μg/ml) to 0.8 OD/ml, and cells were challenged with varying concentrations of STR and NOR for 3 hr at shaking, and 50 μM CCCP was used as a control. 200 μl cell aliquots were transferred to a black wall 96-well plate (Costar). Fluorescence was recorded in a microplate reader (BioTek Synergy H1) for Peredox at 410 nm excitation and 510 nm emission, while mCherry fluorescence was recorded at 588 nm excitation and 610 nm emission. The ratio of fluorescence acquired at the excitation of 410 nm and 510 nm (410/588 nm) were calculated and plotted.

## Measurement of ATP using the PHR-mCherry biosensor

The PHR/mCherry biosensor indicating ATP/ADP ratio (*Akela and Kumar, 2021*) was electroporated in *M. smegmatis*. Cells were grown with kanamycin (20 μg/ml) to 0.8 $OD_{600nm}$/ml, and cells were challenged with varying concentrations of STR and NOR for 3 hr at shaking, and 50 μM CCCP was used as a control. After 3 hr, 100 μl of cells were aliquoted and diluted with 1:10 volume of PBS-0.05% Tween-80 and analysed by flow cytometry (Beckman Coulter CytoFLEX LX). Mean fluorescence intensities (MFI) for PHR acquired at 488 nm excitation and 525 nm emission (B525-FITC-A) was divided by that of the mCherry acquired at 588 nm excitation and 610 nm emission (Y610-mCherry-A). The ratio of fluorescence acquired at the excitation of 410 nm and 510 nm (488/588 nm) were calculated and plotted.

## Detection of ROS using CellROX Deep Red dye

*M. smegmatis* cells were challenged with increasing concentrations of norfloxacin and streptomycin and seeded in a black wall 96-well plate (Costar). After 3 hr of treatments, cells were treated with 5 μM of CellROX Deep Red dye and incubated in the dark at 37 °C for 40 min. Fluorescence was recorded at 645Ex/665Em (nm) in a microplate reader (BioTek Synergy H1).

## Measurement of mutation frequency

Mutation frequency was determined to check if the treatment with the sub-lethal (¼x-$MBC_{99}$ concentrations of norfloxacin and streptomycin, individually) antibiotics increased mutation frequency of cells. 10 ml of exponentially growing cultures of Msm were treated with and without ¼x-$MBC_{99}$ concentrations of norfloxacin and streptomycin for a total of 25 hr at shaking. At 25th hr, 50 μl of cells were serially diluted and plated on LB agar to count CFU/ml, and the rest of the cultures were harvested at 6000 g and resuspended in 300 μl of 7H9+0.05% Tween-80. 300 μl of suspension was plated onto 2 LB agar supplemented with 100 μg/ml (5x-MIC of RIF on agar) of Rifampicin. Plates were incubated in the dark for 4–5 days at 37 °C, and colonies were counted. The colonies that appeared on LB-RIF plates were RIF resisters (Rif$^r$). Mutation frequency was calculated by dividing the number of RIF resistors with the total number of CFUs plated on the LB-RIF plate and compared between control and drug treatment.

## $^{13}$C isotopologue analysis

### Metabolite extraction

The metabolites were extracted and analysed as described elsewhere (*Thomson et al., 2022*). Briefly, Msm cells were grown till exponential phase (0.6 $OD_{600nm}$/ml) and diluted to the inoculum of 0.0025

$OD_{600nm}$/ml in 7h9 supplemented with 0.05% Tween-80, 1% w/v $^{12}C_6$-Glucose and 1% w/v U-$^{13}C_6$ Glucose (50% labelled glucose). Identical to the growth curve and proteomics experiments, antibiotics were added at 2.5 hr and cells were grown with and without ¼x-MBC$_{99}$ concentrations of norfloxacin and streptomycin for a total of 25 hr with shaking. At 25th hr, cells were harvested and washed twice using chilled PBS at 4000 rpm for 10 min. Cells were subsequently resuspended in a pre-chilled methanol:acetonitrile:water (2:2:1), and metabolites were extracted by mechanical lysis with 0.1 mm zirconia beads using FastPrep system (MP Bio) set at 2 cycles of bead beating at 6.5 m/s for 20 s. Lysate was spun at 14,000 rpm for 2 min in a refrigerated centrifuge. The supernatant was filtered using 0.2 µM cellulose acetate filters and stored at –80 degrees until further use.

## LC-MS analysis

The metabolites were separated using the Agilent InfinityLab Poroshell 120 HILIC-Z (2.1×100 mm, 2.7 µm (p/n 675775–924)) column, suitable for polar acidic metabolites. Chromatographic separation was performed using the mobile phase A (10 mM ammonium acetate in water at pH = 9) and mobile phase B (1.0 mM ammonium acetate at pH = 9) in 10:90 (v:v) water/acetonitrile. Both the solvents were supplemented with a 5 µM Agilent InfinityLab deactivator additive (p/n 5191–4506). The gradient used for the separation is as follows; flow rate of 0.5 ml/min: 0 min, 100% B; 0–11.5 min, 70% B; 11.5–15 min, 100% B; 12–15 min, 100% B; and 5 min of re-equilibration at 100% B. Accurate mass measurements were performed by the Agilent Accurate Mass 6545 Q TOF apparatus, connected in line with an Agilent 1290 Infinity II UHPLC. The mass measurement was performed in a negative-ion mode using the following parameters: capillary voltage, 3000 V; fragmentor voltage, 115 V, in centroid 4 GHz mode. The instrument offered mass measurement at less than 5 ppm error with a mass resolution ranging from 10,000–45,000 over the fixed m/z range of 121–955 atomic mass units. Metabolites were identified based on their accurate masses and their accompanying isotopes. The percent $^{13}C$ labelling of each metabolite was determined by dividing the summed peak height ion intensities of all $^{13}C$-labelled species by the total ion intensity of both labelled and unlabelled ionic species using the Agilent Profinder version B.8.0.00 service pack 3.

## Statistical analysis

All the experiments were performed independently at least thrice or more. All the data are represented as mean and the error bars represent the standard deviation. All statistical analyses were performed in GraphPad Prism 9. For the calculation of statistical significance, the Student's t-test (unpaired, two-tailed) was performed in a pair-wise manner (between control and the desired experiment), as indicated in the figures.

## Acknowledgements

We thank Profs. John Mckinney and Vikas Jain, IISER Bhopal for providing us with the gene deletion mutants of *M. smegmatis* of *icl1-2* and *atpD*, respectively. We thank Dr. Areejit Samal for insightful discussions. The authors thank Drs. Amit Singh and Ashwani Kumar for gifting us Mrx1-roGFP2 biosensor, and Peredox-mCherry and PHR-mCherry biosensors, respectively. The authors thank Annesha Adhikari and Rohit Satardekar for assistance in the experiments and Akanksha Agrawal for assistance with creating images. TL thanks CSIR, India for the doctoral research fellowship and the EMBO short-term scientific exchange fellowship. RM thanks DBT, Govt. of India for research funding (BT/PR20820/MED/30/1875/2017). DB thanks the Biotechnology and Biological Sciences Research Council (BBSRC) (BB/T007648/1) for research funding.

## Additional information

### Funding

| Funder | Grant reference number | Author |
|---|---|---|
| Dept. of Biotechnology, Govt of India | BT/PR20820/ MED/30/1875/2017 | Raju Mukherjee |

| Funder | Grant reference number | Author |
|---|---|---|
| Biotechnology and Biological Sciences Research Council | BB/T007648/1 | Dany JV Beste |

The funders had no role in study design, data collection and interpretation, or the decision to submit the work for publication.

## Author contributions

Tejan Lodhiya, Conceptualization, Data curation, Formal analysis, Investigation, Methodology, Writing – original draft; Aseem Palande, Anjali Veeram, Gerald J Larrouy-Maumus, Investigation, Methodology; Dany JV Beste, Resources, Formal analysis, Supervision, Funding acquisition, Methodology, Writing – review and editing; Raju Mukherjee, Conceptualization, Supervision, Funding acquisition, Investigation, Methodology, Project administration, Writing – review and editing

## Author ORCIDs

Tejan Lodhiya ⓘ https://orcid.org/0000-0003-4961-1833
Gerald J Larrouy-Maumus ⓘ https://orcid.org/0000-0001-6614-8698
Raju Mukherjee ⓘ https://orcid.org/0000-0003-2035-4407

Reviewer #1 (Public review): https://doi.org/10.7554/eLife.99656.4.sa1
Reviewer #2 (Public review): https://doi.org/10.7554/eLife.99656.4.sa2
Author response https://doi.org/10.7554/eLife.99656.4.sa3

# Additional files

## Supplementary files

MDAR checklist

## Data availability

All data generated or analysed during this study are included in the supporting files and in the publicly accessible repository under the following entries: MassIVE MSV000099259, MSV000098909, MSV000098929, MSV000098432, MSV000098761, MSV000098771.

The following datasets were generated:

| Author(s) | Year | Dataset title | Dataset URL | Database and Identifier |
|---|---|---|---|---|
| Lodhiya T, Palande A, Veeram A, Larrouy-Maumus GJ, Beste DJV, Mukherjee R | 2025 | Label free proteomics in *M. smegmatis* upon exposure to sub-inhibitory concentration of antibiotics_Norfloxacin_7.5hrs | https://doi.org/10.25345/C5TM72D7D | MassIVE, 10.25345/C5TM72D7D |
| Lodhiya T, Palande A, Veeram A, Larrouy-Maumus GJ, Beste DJV, Mukherjee R | 2025 | Label free proteomics in *M. smegmatis* upon exposure to sub-inhibitory concentration of antibiotics_Norfloxacin_15hrs | https://doi.org/10.25345/C59G5GS61 | MassIVE, 10.25345/C59G5GS61 |
| Lodhiya T, Palande A, Veeram A, Larrouy-Maumus GJ, Beste DJV, Mukherjee R | 2025 | Label free proteomics in *M. smegmatis* upon exposure to sub-inhibitory concentration of antibiotics_Norfloxacin_25hrs | https://doi.org/10.25345/C5QJ78B2Z | MassIVE, 10.25345/C5QJ78B2Z |

*Continued on next page*

*Continued*

| Author(s) | Year | Dataset title | Dataset URL | Database and Identifier |
|---|---|---|---|---|
| Lodhiya T, Palande A, Veeram A, Larrouy-Maumus GJ, Beste DJV, Mukherjee R | 2025 | Label free proteomics in *M. smegmatis* upon exposure to sub-inhibitory concentration of antibiotics_Streptomycin_7.5hrs | https://doi.org/10.25345/C5JD4Q254 | MassIVE, 10.25345/C5JD4Q254 |
| Lodhiya T, Palande A, Veeram A, Larrouy-Maumus GJ, Beste DJV, Mukherjee R | 2025 | Label free proteomics in *M. smegmatis* upon exposure to sub-inhibitory concentration of antibiotics_Streptomycin_15hrs | https://doi.org/10.25345/C5T727V26 | MassIVE, 10.25345/C5T727V26 |
| Lodhiya T, Palande A, Veeram A, Larrouy-Maumus G, Beste DJV, Mukherjee R | 2025 | Label free proteomics in *M. smegmatis* upon exposure to sub-inhibitory concentration of antibiotics_Streptomycin_25hrs | https://doi.org/10.25345/C5HQ3SB5Z | MassIVE, 10.25345/C5HQ3SB5Z |

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
