## [Editor Report · eLife Assessment]

In this **important** work, Lodhiya et al. provide evidence that excessive ATP underlies the killing of the model organism *Mycobacterium smegmatis* by two mechanistically-distinct antibiotics. The data are generally **solid** as the authors deploy multiple, orthogonal readouts and methods for manipulating reactive oxygen species and ATP. The work will be of interest to those studying antibiotic mechanisms of action.

---

## [Referee Report · Reviewer #1 (Public review)]

Summary:

Lodhiya et al. demonstrate that antibiotics with distinct mechanisms of action, norfloxacin and streptomycin, cause similar metabolic dysfunction in the model organism *Mycobacterium smegmatis*. This includes enhanced flux through the TCA cycle and respiration as well as a build-up of reactive oxygen species (ROS) and ATP. Genetic and/or pharmacologic depression of ROS or ATP levels protect *M. smegmatis* from norfloxacin and streptomycin killing. Because ATP depression is protective, but in some cases does not depress ROS, the authors surmise that excessive ATP is the primary mechanism by which norfloxacin and streptomycin kill *M. smegmatis*. In general, the experiments are carefully executed; alternative hypotheses are discussed and considered; the data are contextualized within the existing literature.

Strengths:

The authors tackle a problem that is both biologically interesting and medically impactful, namely, the mechanism of antibiotic-induced cell death.

Experiments are carefully executed, for example, numerous dose- and time-dependency studies; multiple, orthogonal readouts for ROS; and several methods for pharmacological and genetic depletion of ATP.

There has been a lot of excitement and controversy in the field, and the authors do a nice job of situating their work in this larger context.

Inherent limitations to some of their approaches are acknowledged and discussed e.g., normalizing ATP levels to viable counts of bacteria.

Weaknesses:

All of the experiments performed here were in the model organism *M. smegmatis*. As the authors point out, the extent to which these findings apply to other organisms (most notably, slow-growing pathogens like *M. tuberculosis*) is to be determined.

At first glance, some of the results in the manuscript seem to conflict with what has been previously reported in the (referenced) literature. In their response to reviewers, the authors addressed these concerns. Ideally they would have addressed them in the main manuscript too.

Figs. 9 and 10A-B and associated text make the manuscript significantly longer and more descriptive. They are more appropriate to the beginning of a new story rather than the end of the current one.

---

## [Referee Report · Reviewer #2 (Public review)]

Summary:

The authors are trying to test the hypothesis that ATP bursts are the predominant driver of antibiotic lethality of Mycobacteria

Strengths:

No significant strengths in the current state as it is written.

Weaknesses:

A major weakness is that *M. smegmatis* has a doubling time of three hours and the authors are trying to conclude that their data would reflect the physiology of M. tuberculossi which has a doubling time of 24 hours. Moreover, the authors try to compare OD measurements with CFU counts and thus observe great variabilities.

Comments on revisions:

The authors confirm they are using CFU counts, but then Figure 1 has 0 as the first data point on the Y-axis. This should be somewhere between 10e5 or 10e6. CFU would not start at 0, your initial inoculum has to be more than 0 to have something to challenge.

---

## [Author Response]

The following is the authors’ response to the previous reviews.

**Reviewer #1 (Public review):**
Summary:Lodhiya et al. demonstrate that antibiotics with distinct mechanisms of action, norfloxacin and streptomycin, cause similar metabolic dysfunction in the model organism *Mycobacterium smegmatis*. This includes enhanced flux through the TCA cycle and respiration as well as a build-up of reactive oxygen species (ROS) and ATP. Genetic and/or pharmacologic depression of ROS or ATP levels protect *M. smegmatis* from norfloxacin and streptomycin killing. Because ATP depression is protective, but in some cases does not depress ROS, the authors surmise that excessive ATP is the primary mechanism by which norfloxacin and streptomycin kill *M. smegmatis*. In general, the experiments are carefully executed; alternative hypotheses are discussed and considered; the data are contextualized within the existing literature.

We thank the reviewer for the very comprehensive summary of the study.

Strengths:The authors tackle a problem that is both biologically interesting and medically impactful, namely, the mechanism of antibiotic-induced cell death.Experiments are carefully executed, for example, numerous dose- and time-dependency studies; multiple, orthogonal readouts for ROS; and several methods for pharmacological and genetic depletion of ATP.There has been a lot of excitement and controversy in the field, and the authors do a nice job of situating their work in this larger context.Inherent limitations to some of their approaches are acknowledged and discussed e.g., normalizing ATP levels to viable counts of bacteria.

We thank the reviewer for the encouraging comments.

Weaknesses:All of the experiments performed here were in the model organism *M. smegmatis*. As the authors point out, the extent to which these findings apply to other organisms (most notably, slow-growing pathogens like *M. tuberculosis*) is to be determined. To avoid the perception of overreach, I would recommend substituting "*M. smegmatis*" for Mycobacteria (especially in the title and abstract).At first glance, a few of the results in the manuscript seem to conflict with what has been previously reported in the (referenced) literature. In their response to reviewers, the authors addressed my concerns. It would also be ideal to include a few lines in the manuscript briefly addressing these points. (Other readers may have similar concerns).In the first round of review, I suggested that the authors consider removing Figs. 9 and 10A-B as I believe they distract from the main point of the paper and appear to be the beginning of a new story rather than the end of the current one. I still hold this opinion. However, one of the strengths of the eLife model is that we can agree to disagree.

We acknowledge the reviewer’s concern and have changed title of the manuscript by including *Mycobacterium smegmatis* instead of Mycobacteria. The abstract already mentioned the same.

In the discussion section of the revised manuscript, we have already addressed and analysed our results extensively within the context of the available literature, regardless of whether our findings aligned with or differed from previous studies. We still believe that the mentioned discussion will help suffice to explain our results to the readers.

In this manuscript we also sought to assess the bacteria's ability to counteract drug induced stresses, contributing to our understanding of how antibiotic tolerance develop in *Mycobacterium smegmatis*. Results presented in Figure 9 clearly demonstrate that *M. smegmatis* attempt to reduce respiration by decreasing flux through the complete TCA cycle, thereby mitigating ROS and ATP production in response to antibiotics. Additionally, the bacterial response also included increased expression of the protein Eis, which is exemplar for intrinsic drug resistance, with a concomitant increase in mutation frequency, thereby hinting at the development of antibiotic tolerance followed by resistance. We still believe that these data should be included to support our observations and they make the study more comprehensive.

**Reviewer #2 (Public review):**
Summary:The authors are trying to test the hypothesis that ATP bursts are the predominant driver of antibiotic lethality of MycobacteriaStrengths:No significant strengths in the current state as it is written.Weaknesses:A major weakness is that *M. smegmatis* has a doubling time of three hours and the authors are trying to conclude that their data would reflect the physiology of M. tuberculossi that has a doubling time of 24 hours. Moreover, the authors try to compare OD measurements with CFU counts and thus observe great variabilities.Comments on revisions:I am surprised that the authors simply did not repeat the study in figure one with CFU counts and repeated in triplicate. Since this is *M. smegmatis*, it would take no longer than two weeks to repeat this experiment and replace the figure. I understand that obtaining CFU counts is much more laborious than OD measurements but it is necessary. Your graph still says that there is 0 bacteria at time 0, yet in your legend it says you started with 600,000 CFU/ml. I don't understand why this experiment was not repeated with CFU counts measured throughout. This is not a big ask since this is *M. smegmatis* but it appears that the authors do not want to repeat this experiment. Minimally, fix the graph to represent the CFU.

We acknowledge the reviewer’s concern and have changed title of the manuscript by specifying *Mycobacterium smegmatis* instead of Mycobacteria.

It is still not clear to the authors what the reviewer mean by OD measurements. All the data presented in the entire manuscript , including in Figure 1 are solely based on CFU measurements. So, as suggested by the reviewer, all experiments are already presented in terms of CFU.